# Strongly coupled interface ferroelectricity and interface superconductivity in amorphous LaAlO₃/KTaO₃(111)

M. D. Dong[1,2,3,5], X. B. Cheng[1,2,3,5], M. Zhang [4,5] & J. Wu [1,2,3]✉

Interfaces can differ from their parent compounds in charge, spin, and orbital orders, providing fertile ground for emergent phenomena, strongly correlated physics, and novel device applications. Here, we present evidence of a ferroelectric order at the interface of two oxides, amorphous LaAlO₃/KTaO₃(111), where two seemingly mutually exclusive orders—ferroelectricity and superconductivity—coexist. Ferroelectricity is confirmed through scanning transmission electron microscopy (STEM), second harmonic generation (SHG) microscopy, and piezoelectric force microscopy (PFM). STEM reveals a displacement of K atoms relative to Ta atoms, facilitated by oxygen vacancies at the LAO/KTO interface. The resulting ferroelectric polarization is locally switchable by applying a voltage between the PFM tip and the LaAlO₃ film. Flipping ferroelectric polarization reduces interfacial conductivity by more than 1000 times, simultaneously suppresses superconductivity. Moreover, the ferroelectric hysteresis correlates with hysteretic changes in interfacial conductivity and the superconducting transition temperature ($T_c$), providing clear evidence of coupling between ferroelectricity and superconductivity. These findings open a pathway to ferroelectric superconductivity with broken inversion symmetry and non-volatile control of superconductivity.

To overcome the hurdle that the screening of itinerant electrons effectively suppresses the long-range Coulomb forces between electric dipoles and consequently the ferroelectric order, Anderson and Blount proposed in 1964 that ferroelectric metal might exist if itinerant charges are weakly coupled to transverse optical phonons[1]. Since then, experimental evidence of this novel state remained elusive until the ferroelectric metal, WTe₂, with a switchable polarization axis, was discovered[2,3]. So far, the coexistence of electric polarization and metallic conductivity[4] has been observed in several materials, including perovskite oxides[5–7], layered perovskites[8], and two-dimensional materials[2,3,9], although the polarization switchability for many of them has yet to be demonstrated.

The lower dimensionality and carrier density reduce electron screening and thus are beneficial for stabilization of ferroelectric order. Besides two-dimensional materials, interfaces of heterostructures, which host a two-dimensional electron gas[10,11], represent another conducting material system with lower dimensionality and carrier density. It remains an intriguing question whether ferroelectric order might also emerge in these systems, a topic that warrants in-depth theoretical and experimental investigation. Even more attractively, at low temperatures, interface superconductivity has been discovered in heterointerfaces such as La₂₋ₓSrₓCuO₄/La₂CuO₄[12–14], LaAlO₃/SrTiO₃[15,16], monolayer FeSe/SrTiO₃[17], EuO/KTO[18,19], LAO/KTO[16,20,21], and other KTO-based heterointerfaces[22–27]. Here, SrTiO₃ and KTO are both

[1]Department of Physics, School of Science, Westlake University, Hangzhou 310024, China. [2]Research Center for Industries of the Future, Westlake University, Hangzhou 310024, China. [3]Key Laboratory for Quantum Materials of Zhejiang Province, School of Science, Westlake University, Hangzhou 310024, China. [4]School of Physics, Zhejiang University, Hangzhou 310027, China. [5]These authors contributed equally: M. D. Dong, X. B. Cheng, M. Zhang ✉e-mail: wujie@westlake.edu.cn

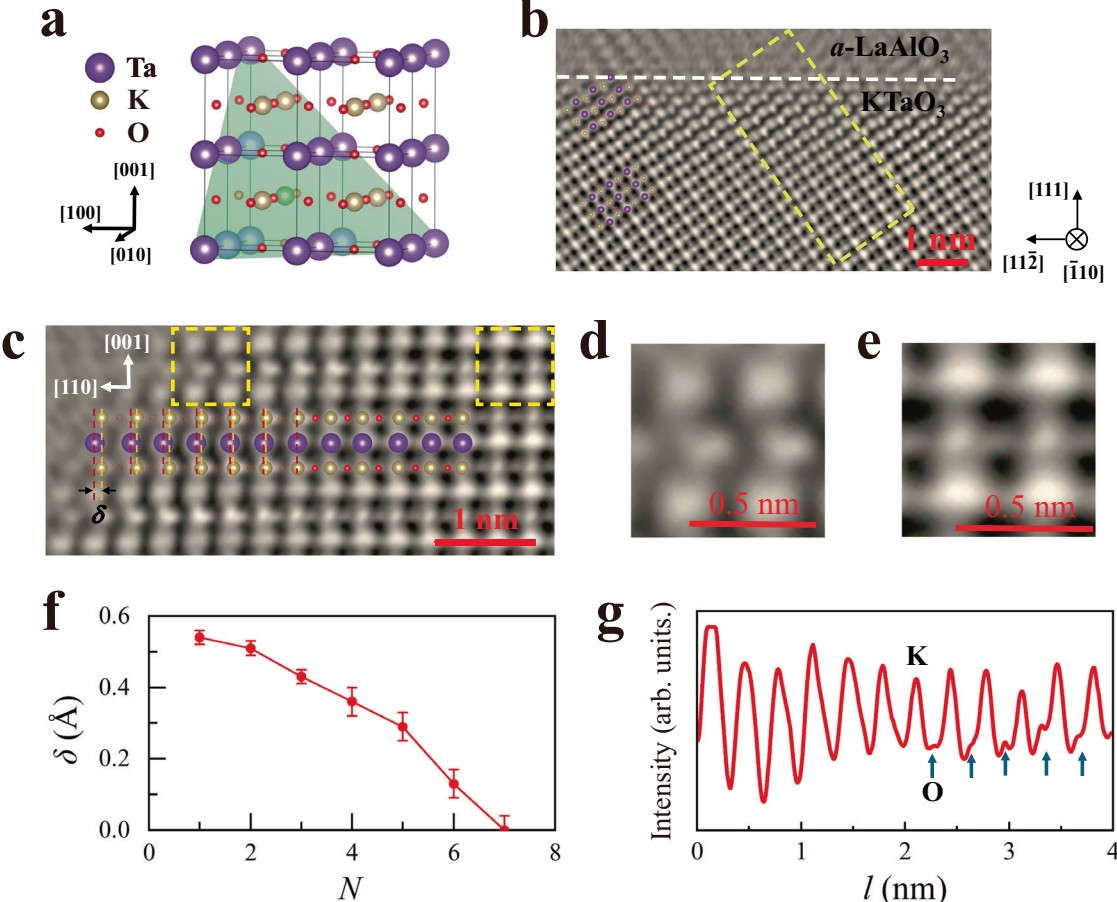

**Fig. 1 | Atomic displacement and oxygen vacancies at the LAO/KTO interface.**
**a** A schematic drawing of the KTO crystalline structure. **b** The iDPC-STEM image
with atomic resolution. The dashed line denotes the boundary between LAO and
KTO. **c** The amplified view of atoms in the yellow box of panel (**b**). The relative
displacement between K and Ta atoms is denoted by $\delta$. $\delta$ is substantial (-0.54 Å)
right at the LAO/KTO interface and quickly approaches zero for atoms distant from
the interface. Concomitantly, some oxygen atoms are clearly missing at the
interface (open circles). **d**, **e** The amplified view of atoms in boxes 1 and 2 of panel
(**c**), respectively. The atomic structures close to and far away from the interface
show remarkable differences. **f** The measured $\delta$ as a function of $N$, where $N$ is the
number of Ta−O plane counting from the interface. **g** The line profile (corre-
sponding to the purple dashed line in panel (**c**)) shows the locations of oxygen
atoms, which are missing in the vicinity of the interface.

dielectric materials with large dielectric constants (-20,000 and 4000
at low temperature[28,29], respectively), indicating that they are on the
verge of a paraelectric to ferroelectric phase transition. However,
signatures of ferroelectricity in $SrTiO_3$ or KTO-based interfaces remain
elusive till now. Ferroelectric polarization at the interface, if existing or
induced, may be intertwined with superconductivity. Ferroelectric
superconductors are exceptionally rare in single-phase materials, with
the only known examples being $Sr_{1-x}Ca_xTiO_{3-\delta}$ bulk[30] and bilayer
$MoTe_2$[31]. In contrast, heterointerfaces offer infinite combinations of
parent compounds, potentially opening the door to a new family of
ferroelectric superconductors.

We deposited amorphous $LaAlO_3$ films onto $KTaO_3$(111) substrates
for the study of interface ferroelectricity. For simplicity and read-
ability, the abbreviation "LAO/KTO" is used throughout the paper for
"amorphous $LaAlO_3/KTaO_3$". Both LAO and KTO are insulators, and the
electrical conductivity of LAO/KTO(111) comes entirely from its inter-
face. Superconductivity with a transition temperature of approxi-
mately 1.7 K resides on the KTO side of the interface[18,20,24], and the
thickness of the (super)conducting layer is about 4 nm[18,20], which is less
than the coherence length of 18.8 nm[18,20], illustrating that the super-
conductivity is two-dimensional. KTO-based interface super-
conductivity shows an intriguing dependence on substrate
orientation; for example, LAO/KTO(111) and LAO/KTO(110) are super-
conducting, while LAO/KTO(100) is not[18,19,21]. This is attributed to

pairings mediated by transverse optical phonons[26]. Superconductivity
in LAO/KTO(111) can be controlled by back-side gating the KTO(111)
substrate, illustrating the dominant role of carrier mobility and inter-
facial disorder[20]. Thus, the properties at the interface, such as lattice,
phonons, carrier density, and mobility, are crucial for understanding
interface superconductivity.

## Results and discussion

A 10 nm LAO film was deposited onto a preconditioned KTO(111)
substrate by pulsed laser deposition. During deposition, the substrate
was stabilized at 300 °C and the oxygen partial pressure was kept at
$1 \times 10^{-5}$ mbar. Water vapor with a partial pressure $1 \times 10^{-7}$ mbar was
added to the growth chamber, a method that has been proven to
improve film quality and enhance the superconducting $T_c$. The details
of the synthesis are included in the "Methods" section, and the growth
recipe has been continuously optimized through many rounds of
synthesis-characterization cycles.

The integrated differential phase contrast (iDPC) STEM images of
the LAO/KTO(111) heterointerface clearly show the coexistence of K
atom displacement and oxygen vacancies right at the interface (Fig. 1).
The top LAO film is amorphous, lacking spatial periodicity, in contrast
to the regular lattice formed by K, Ta, and O atoms in the KTO sub-
strate (Fig. 1b). The cation interdiffusion across the interface is limited
to within 1 nm, as determined by energy-dispersive spectroscopy (EDS)

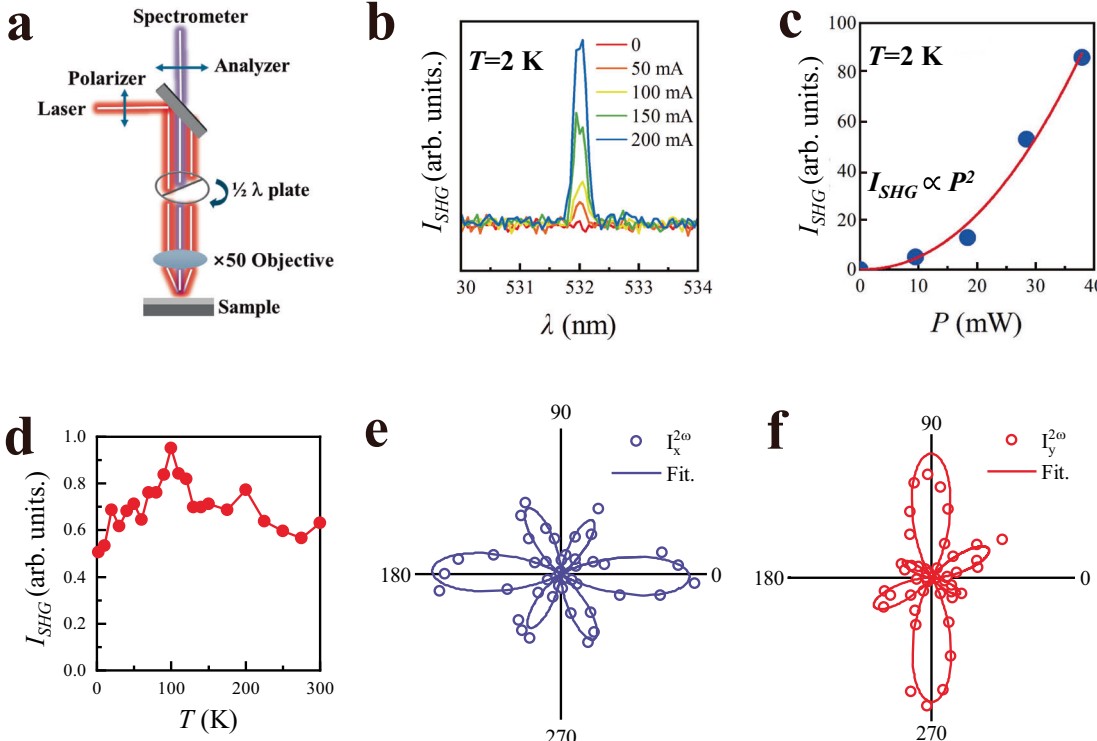

**Fig. 2 | Ferroelectric polarization of the LAO/KTO interface characterized by the SHG signal. a** A schematic drawing of the SHG microscope. The half-wavelength plate can be continuously rotated to measure both $I_x^{2\omega}(\phi)$ and $I_y^{2\omega}(\phi)$, where $\phi$ is the angle between the polarization of the incident light and the KTO crystallographic $[1\bar{1}0]$ direction. **b** The wavelength of the incident laser is 1064 nm, and a pronounced SHG signal peaked at 532 nm wavelength is manifested in the reflected beam from LAO/KTO(111) at $T = 2$ K. **c** The intensity of the SHG signal $I_{SHG}$ is proportional to $P^2$, where $P$ is the power of the incident laser. **d** $I_{SHG}$ persists at low temperatures. **e**, **f** $I_x^{2\omega}(\phi)$ and $I_y^{2\omega}(\phi)$ measured at 300 K, respectively. The data (open circles) are well fitted (solid curves) by the expressions based on the $m$ polar group.

mapping of the La, Al, K, and Ta atoms (see Fig. S1 of Supplementary Information for details).

K atoms in the vicinity of the LAO/KTO interface are displaced relative to the lattice center formed by Ta and O atoms (Fig. 1c–e). The K displacement, $\delta$, is along the KTO [110] direction and depends sensitively on its distance from the interface. In contrast, the iDPC-STEM image taken from the KTO $(11\bar{2})$ plane (Fig. S2 of Supplementary Information), which is perpendicular to the KTO $(\bar{1}10)$ plane in Fig. 1, shows no appreciable K displacement along the KTO $[\bar{1}10]$ direction. This verifies that the K displacement is precisely along the KTO [110] direction. The retrieved $\delta$ ($N$) relation (Fig. 1f) shows that $\delta$ is huge (-0.54 Å) at the interface and decreases rapidly with $N$, where $N$ is the number of TaO$_2$ planes along the KTO [001] direction counting from the interface. The identified positions of oxygen atoms show that a significant number of oxygen atoms in the K–O plane are missing in the KTO lattice close to the LAO/KTO interface, contrasted by the KTO lattice far away from the interface (Fig. 1c–e). This is corroborated by the change in electron energy loss spectroscopy (EELS) spectra on oxygen K edge taken from areas close and far away from the LAO/KTO interface (Fig. S3 of Supplementary Information). From the line profile (Fig. 1g), the KTO layers accommodating oxygen vacancies range from $N = 1$ to $N = 6$. Thus, the critical thickness at which oxygen atoms are restored to the lattice coincides with the critical thickness at which $\delta$ diminishes (Fig. 1f). This clearly demonstrates that the K displacement is associated with oxygen vacancy.

It should be mentioned that exposure to a high-energy electron beam quickly degrades LAO/KTO samples, especially near the interface, which is susceptible to beam damage, as oxygen vacancies may be created, moved, or removed during STEM imaging. Thus, acquiring atom-resolved images within a very limited time frame is challenging.

The STEM images shown in this work are all taken from as-grown samples before severe beam damage occurs.

The ferroelectric polarization and inversion-symmetry breaking originate from the displacement of the K atom and the presence of oxygen vacancies that give rise to the nonlinear optical effect detected by the SHG microscopy (Fig. 2a). With both the incident laser and the generated second harmonic light normal to the film surface, and their polarizations in-plane, the SHG signal is sensitive only to the in-plane component of electric polarization. Since charge transfer between LAO and KTO produces an out-of-plane electric polarization, the SHG signal primarily originates from the in-plane component of ferroelectric polarization (Fig. 1). At 2 K–below the onset temperature of superconductivity and the lowest temperature accessible to our setup –a linearly polarized 1064 nm laser incident normally on the LAO/KTO(111) sample generates a strong second-harmonic signal at 532 nm (Fig. 2b). This signal is absent in the KTO(111) substrate alone (Fig. S5 of Supplementary Information). The SHG intensity scales quadratically[32] with the laser power $P$, $I_{SHG} \propto P^2$ (Fig. 2c). Temperature-dependent measurements of $I_{SHG}(T)$ show no abrupt transition as $T$ increases from 2 K to 300 K (Fig. 2d), indicating that ferroelectric order persists to room temperature. At 300 K, rotating the half-wavelength plate allows in-plane rotation of the incident light polarization. The measured SHG intensity, $I_x^{2\omega}$ and $I_y^{2\omega}$, with polarization parallel and perpendicular to that of the incident light, respectively, show angular dependence on $\phi$ (Fig. 2e, f), where $\phi$ denotes the angle between the polarization of the incident light and the KTO crystallographic $[1\bar{1}0]$ direction. Both $I_x^{2\omega}(\phi)$ and $I_y^{2\omega}(\phi)$ display 6 peaks evenly spaced by ~60° over 360°, reflecting the symmetry of the KTO (111) surface. The data are simultaneously fitted by expressions based on $m$ polar group, yielding a ferroelectric polarization oriented along the KTO [110] direction, in agreement with

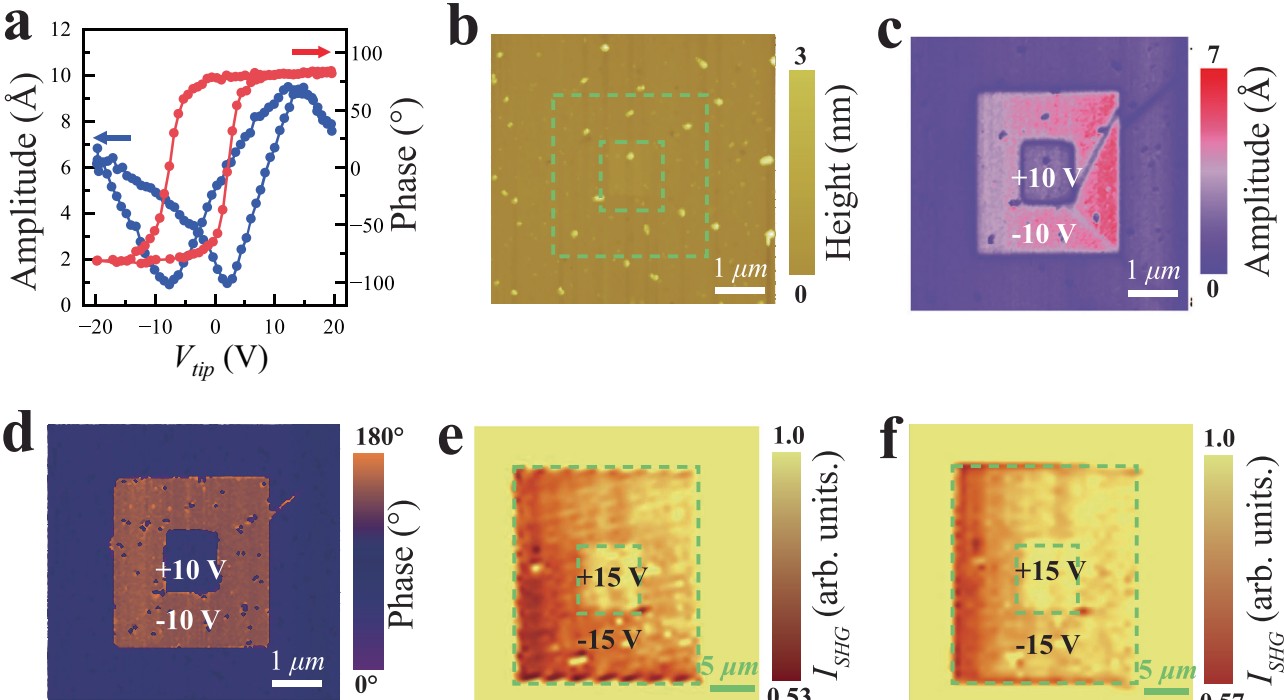

**Fig. 3 | Ferroelectric domains created by PFM. a** The amplitude *A* and phase *P* of the piezoelectric response by ramping the voltage $V_{tip}$ applied between the PFM tip and the LAO/KTO sample. **b** The atomic force microscopy shows a flat surface of LAO/KTO. **c**, **d** The amplitude and phase of PFM manifest clear contrast for different areas taken after writing ferroelectric domains, evidencing the switchability of ferroelectricity. A ±10 V voltage is applied on the PFM tip during scanning to locally flip ferroelectric domains. The $4 \times 4\,\mu m^2$ area within the yellow dashed lines was scanned with −10 V voltage on the PFM tip, and then the inner $2 \times 2\,\mu m^2$ area within the dashed red lines was scanned with +10 V voltage. **e**, **f** The SHG images acquired within 4 h (**e**) and after 70 h (**f**) after pattern writing. The persistent contrast after 70 h indicates a stable, switchable ferroelectric order in LAO/KTO(111).

STEM results (see the Methods section for details). To unambiguously demonstrate ferroelectricity in the superconducting state, we further replaced the picosecond laser with a femtosecond laser and lowered the laser power to only 1.5 mW for the measurements, compared with 83 mW in Fig. 2e, f. With these improvements, the base temperature of the LAO/KTO(111) film reaches 1.62 K—well below its superconducting temperature. Despite this drastically reduced laser intensity, the SHG signal remains significant and approximately temperature-independent (Fig. S12 of Supplementary Information), confirming that ferroelectric polarization is robust from room temperature all the way down to the superconducting regime.

The switchability of ferroelectric polarization is investigated using the PFM method. On a flat area of LAO/KTO(111), in response to the voltage applied to the PFM tip, we observed a "butterfly" curve in the switching-spectroscopy PFM (SS-PFM) amplitude, along with a hysteresis loop in the PFM phase, which are both characteristic signatures of ferroelectric order[33–35] (Fig. 3a). It should be emphasized that the application of PFM tip voltage and measurements were carried out in a closed chamber filled with $N_2$ gas. This removes water vapor from environments that breaks down under applied tip voltage and introduces $H^+$ or $OH^−$ ions into the sample[36,37]. Moreover, any ions induced would have to migrate through the 10 nm-thick LAO layer to reach the interface, making their influence even less likely. In this way, we eliminate the effects due to ion implantation and the measured ferroelectric signal is intrinsic to the LAO/KTO(111) heterostructure.

No ferroelectric domain has been found in the as-grown LAO/KTO(111) under PFM imaging, consistent with the absence of domains in SHG images. However, ferroelectric domains can be created and manipulated by locally switching the ferroelectric polarization with the PFM tip[33–35]. By applying a +10 V voltage on the PFM tip, which is sufficient to flip the ferroelectric polarization (Fig. 3a), a $4 \times 4\,\mu m^2$ area with polarization opposite to the surrounding as-grown area was created, featuring clear domain boundaries (Fig. 3b–d). Within this square, a smaller concentric $2 \times 2\,\mu m^2$ square was created by applying a −10 V voltage on the PFM tip to flip back the ferroelectric polarization. This clearly demonstrates the switchability of ferroelectric polarization. Note that the outside as-grown area has the same contrast as the inner smaller square created by the −10 V tip voltage, implying that the as-grown LAO/KTO(111) is in a single domain state, so no domain contrast is present in the pristine state for PFM and SHG imaging.

Beyond ferroelectric order, redistribution of oxygen vacancies may also contribute to the PFM response, as previously noted in $LaAlO_3$/$SrTiO_3$[38]. To distinguish between these two contributions, we measured the time decay of the PFM domain contrast, which typically decays within hours when induced by oxygen-vacancy redistribution[39], in contrast to the stability of ferroelectric domains arising from intrinsic ferroelectric order. After the concentric square pattern was written by the PFM tip, the PFM amplitude decreased rapidly from 2.7 to 1.5 in the first 1 h, then more gradually from 1.5 to 0.5 between 1 and 12 h, after which it stabilized (see Fig. S7 of Supplementary Information). In stark contrast, the concurrently acquired electrostatic force microscopy (EFM) images showed a rapid contrast decay, vanishing within 12 h. These observations suggest that the decaying components of the PFM and EFM signals likely relate to oxygen-vacancy redistribution, while the persistent PFM signal originates from ferroelectric polarization. Moreover, we obtained SHG images after pattern writing (Fig. 3e), which are sensitive to the in-plane component of ferroelectric polarization and thus complement the PFM measurements. As the out-of-plane component was flipped during pattern writing, the coupled in-plane component was switched simultaneously, producing a contrast between regions written with +15 and −15 V applied to the PFM tip. Remarkably, this SHG contrast persisted after 70 h (Fig. 3f), consistent with the sustained PFM amplitude, thereby confirming its origin in ferroelectric polarization.

For comparison, an LAO/MgO(001) film was grown under conditions identical to the superconducting LAO/KTO(111) samples. Applying ±30 V on the PFM tip—well above the voltage applied for LAO/KTO(111)—we wrote a concentric square pattern on LAO/MgO. While both PFM and EFM contrast appeared, they vanished within 48 h (Fig. S9 of Supplementary Information), indicating that these signals likely arise from surface charging or oxygen-vacancy effects[38–40]. No SHG signal was detected from LAO/MgO, even immediately after pattern writing, indicating that surface charging or oxygen vacancies do not generate SHG signals. These results demonstrate that neither the amorphous LAO film itself nor its interface with MgO produces the PFM and SHG signals observed in LAO/KTO(111).

Combining the results from STEM, SHG, and PFM methods, we conclude that ferroelectricity resides at the interface of LAO/KTO(111), and that this interface ferroelectricity can be switched by an external electric field. Next, we will show the coupling between interface ferroelectricity and interface superconductivity.

We ramp up the PFM tip voltage step-by-step to drive the LAO/KTO through a complete ferroelectric hysteresis loop, while simultaneously measuring the corresponding SHG signal and longitudinal resistance to elaborate on the one-to-one correspondence between ferroelectricity and superconductivity. To do this, a designated area between two voltage contacts on a Hall bar is scanned by the PFM tip at a fixed voltage (Fig. 4a), allowing ferroelectricity in this area to be modulated[41] (Fig. 4b). By ramping down the tip voltage $V_{tip}$ to −21 V, then up to +50 V and eventually back to 0 V, the SHG signal $I_x^{2\omega}(\phi)$ and $I_y^{2\omega}(\phi)$ evolves with $V_{tip}$ accordingly (Fig. 4c, d). Consistent with Fig. 2d, e, $I_x^{2\omega}(\phi)$ and $I_y^{2\omega}(\phi)$ both manifest six peaks with a 60° interval. The SHG intensities at the peaks change with $V_{tip}$, but the peak angles remain the same. Since ferroelectricity has both in-plane and out-of-plane components, and the tip writing only modulates the out-of-plane component, the fittings of $I_x^{2\omega}(\phi)$ and $I_y^{2\omega}(\phi)$ require modeling of sophisticated ferroelectric domain distributions in the intermediate states of the hysteresis loop. Nevertheless, putting these details aside, the evolution of $I_x^{2\omega}(\phi)$ and $I_y^{2\omega}(\phi)$ with $V_{tip}$, corroborated with PFM imaging of ferroelectric domains (Fig. 4b), undoubtedly further rules out the possibility of ion implantation by the tip writing and verifies its effectiveness in modulating ferroelectric domains.

It should be noted that the effect of tip writing decays over time, particularly when $V_{tip}$ < −10 V. Using longitudinal resistance $R$ as an indicator, the effect decayed rapidly within the first minute and then gradually relaxed over several hours until stabilization was reached (inset of Fig. 4e). Because transferring the sample and setting up the optics takes time, SHG measurements were performed 30 min after tip writing, when the initial fast decay phase had already ended. By contrast, the $R$ values in Fig. 4e were measured in situ about 3 s after writing, capturing the maximum effect. For the $R(T)$ measurements shown in Fig. 4f, g, we waited 10 h after writing to ensure the effect had stabilized before initiating cooldown. Apparently, the time decay of $R$ parallels the decay of PFM and SHG signals after writing. Once ferroelectric polarization is flipped, oxygen vacancies redistribute accordingly, a process that requires several hours to stabilize. This redistribution accounts for the decaying components observed in the PFM amplitude and $R$(Time). After the decay phase, the remaining PFM and SHG contrasts, together with the residual change in $R$, reflect intrinsic effects directly associated with ferroelectric polarization switching.

As $V_{tip}$ ramps, the resistance $R$ shows clear hysteresis behavior (Fig. 4e) that is directly linked to the ferroelectric hysteresis loop. The sharp increase or decrease of $R$ at $V_{tip}$ ~ −17 and 6 V, respectively, corresponds to critical voltages to flip ferroelectric polarization. Remarkably, the change in $R$ reaches more than $10^5$ times for two saturated states of the hysteresis loop at 300 K, implying a drastic effect of ferroelectric polarization orientation on interfacial conductivity. More importantly, compared to the pristine LAO/KTO sample with $T_c$ = 1.8 K (Here, $T_c$ is denoted as the midpoint of the superconductivity

transition), superconductivity disappears at $V_{tip}$ = −17 V. As $V_{tip}$ reverses its sign and flips the ferroelectric polarization back to the pristine state at $V_{tip}$ = +50 V, superconductivity reappears (Fig. 4f, g). The writing cycles and consequent modulation of superconductivity are repeatable, which clearly demonstrates the non-volatile modulation of superconductivity by ferroelectricity. The interface ferroelectricity and its strong coupling with interface superconductivity are universal for synthesized LAO/KTO(111) samples. More examples from other samples are shown in Fig. S10 of the Supplementary Information. The coercivity and shape of the hysteresis loop vary slightly from sample to sample, but the conclusions remain the same.

Notably, $R(T)$ for the high-resistance state ($V_{tip}$ = −21 V) has a positive slope from room temperature down to low temperatures, characteristics of a metallic state. However, the calculated mean free path for this state is shorter than the lattice constant, which is unphysical. This presumably suggests that conduction is highly inhomogeneous and dominated by percolative current paths. Consequently, the resistance is high but retains a metallic temperature dependence. In addition, an upturn in $R(T)$ appears below 30 K, likely arising from weak localization due to reduced dimensionality and enhanced disorder scattering in the high-resistance state.

The modulation of (super)conductivity arises from the combined effects of ferroelectric polarization $\vec{P}$, disorder scattering and oxygen vacancies. Firstly, flipping $\vec{P}$ from its pristine state effectively reduces the interfacial electron density. The work-function imbalance between LAO and KTO drives electron transfer from the LAO film to the KTO substrate, forming a two-dimensional electron gas (2DEG). This interfacial charge transfer creates an internal electric field $\vec{E}$ due to electron depletion on the LAO side and accumulation on the KTO side (Fig. 4h). Analogous to a $p-n$ junction, electron accumulation strengthens $\vec{E}$ and suppresses further transfer, with the amount of transferred charge determined by the interfacial chemical-potential balance. $\vec{P}$ at the LAO/KTO interface introduces an additional field. When its out-of-plane component, $\vec{P}_\perp$, is antiparallel to $\vec{E}$, the threshold $\vec{E}$ field for halting transfer increases, yielding higher 2DEG density (Fig. 4i). When parallel, the 2DEG density decreases. In pristine LAO/KTO(111), $\vec{P}_\perp$ is antiparallel to $\vec{E}$ since this is the lowest energy ground state. Therefore, resistance is low in the pristine state and increases significantly as $\vec{P}$ is flipped. Secondly, the modification of disorder scattering affects the electron mobility. The flipping of $\vec{P}$ alters the profile of the potential well confining the 2DEG. Electrons in a narrower potential well experience stronger scattering from disorder, resulting in higher resistance. This is akin to the electric field control of interface superconductivity in LAO/KTO(111) by the backside gating[20], where the modulation of superconductivity is also ascribed mainly to a change in charge mobility due to modification of disorder scattering. Thirdly, redistribution of oxygen vacancies affects both electron density and mobility. Ferroelectric polarization is correlated with oxygen vacancies at the interface (Fig. 1), and the flipping of $\vec{P}$ is inevitably accompanied by the redistribution of oxygen vacancies. This is consistent with literature reports[42,43] that oxygen vacancies can enhance electric polarization or piezoelectricity in perovskite oxide films. Additionally, the effect of electron trapping by these vacancies, which was found in $LaAlO_3/SrTiO_3$[44–46], may also play a role by reducing electron density and increasing resistance. Switching experiments, however, provide evidence against charge trapping as the dominant mechanism (Fig. S13 of Supplementary Information). As this work represents an initial step in understanding ferroelectric superconductivity, it is premature to conclude whether there are additional novel features of ferroelectric superconductivity that may contribute to the modulation of (super)conductivity. This remains an open question for future investigation.

To understand why the pristine ferroelectric state is likely a single domain, symmetry-breaking mechanisms must be considered. In the out-of-plane direction, the electric field $\vec{E}$, generated by charge transfer across the LAO/KTO interface, breaks the up-down symmetry,

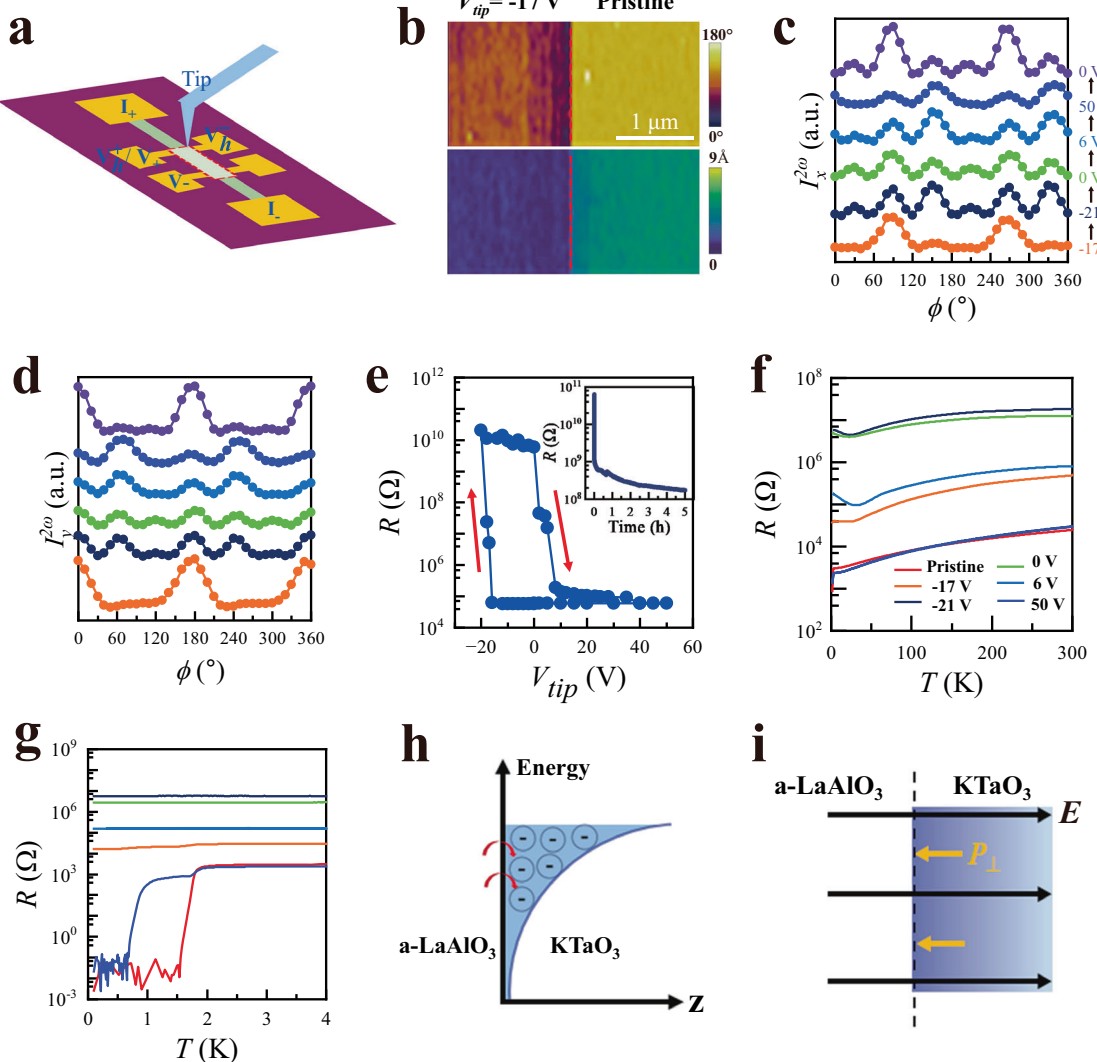

**Fig. 4 | Strong coupling between ferroelectricity and superconductivity. a** A schematic drawing of the experimental setup. The area inside the dashed lines was modulated by PFM tip scanning with variable tip voltage, and then its longitudinal/Hall resistance was measured by a standard four-point measurement scheme. In this way, the correlation between ferroelectric hysteresis and electric transport can be elucidated. **b** The boundary between the written (with $V_{tip} = -17$ V) and pristine area is clearly visible on both the PFM amplitude and phase mapping, verifying that the ferroelectricity has been locally modulated. **c, d** The SHG signal $I_x^{2\omega}(\phi)$ and $I_y^{2\omega}(\phi)$ at 300 K corresponding to the cycling of $V_{tip}$. The sophisticated evolution of $I_x^{2\omega}(\phi)$ and $I_y^{2\omega}(\phi)$ with $V_{tip}$ reflects the motion and flipping of ferroelectric domains. **e** The LAO/KTO longitudinal resistance manifests clear hysteresis behavior in response to PFM tip writings at 300 K. Remarkably, the change in resistance is gigantic and reaches more than $10^5$ times for two saturated states of the hysteresis loop. The inset shows the decay of $R$ as a function of time passed after the writing process. **f** The corresponding $R(T)$ at different $V_{tip}$. $R(T)$ returns to that of the pristine state at the end of the $V_{tip}$ loop after $V_{tip} = 50$ V is applied, verifying that the change is non-volatile. **g** $R(T)$ at low temperatures manifests the demise and reentrance of superconductivity during the cycling of $V_{tip}$, evidencing the strong coupling between superconductivity and ferroelectricity. **h** Schematic drawing of the formation of 2DEG due to charge transfer from amorphous LaAlO₃ (a-LaAlO₃) to KTaO₃, and transferred electrons are confined in a potential well. An internal electric field $\vec{E}$ is created due to electron depletion on the LAO side and accumulation on the KTO side. **i** The out-of-plane component of the ferroelectric polarization, $\vec{P}_\perp$, is antiparallel to $\vec{E}$, for minimization of electrostatic energy. $\vec{P}_\perp$ increase the threshold $\vec{E}$ field for halting charge transfer, yielding higher 2DEG density.

making the lowest energy state for $\vec{P}_\perp$ antiparallel to $\vec{E}$ (Fig. 4i). For the in-plane component $\vec{P}_{//}$, emergent electronic nematicity spontaneously breaks the rotational symmetry and lifts the energy degeneracy of $\vec{P}_{//}$. Supporting evidence for electronic nematicity in LAO/KTO(111) has been reported[27,47], including our recent work[46], where we used the angle-resolved resistivity method to measure anisotropy in electrical transport along various in-plane directions. The results provided compelling evidence for electronic nematicity in the normal, superconducting, fluctuating, and quantum metal states.

The discovery of interface ferroelectricity significantly extends the families of materials hosting ferroelectricity, and its easy tunability due to lower dimensionality is beneficial for device applications. The coupling between interface ferroelectricity and interface superconductivity engenders non-volatile modulation of superconductivity and fuels research on superconductivity without inversion symmetry.

## Methods

### Film synthesis and device fabrication

A 248 nm KrF excimer laser was used with a 0.5–0.7 J cm⁻² laser fluence and 10 Hz laser repetition rate to deposit 10 nm LAO films onto KTO (111) substrates by PLD. The sample temperature was maintained at 300 °C during growth, in a mixed atmosphere of $1 \times 10^{-5}$ mbar O₂ and $1 \times 10^{-7}$ mbar H₂O vapor. For the Hall bar used for electrical transport measurements in Fig. 4, we employed standard UV-lithography and lift-off techniques to pre-pattern the KTO substrate and then deposit LAO films to form the desired device.

## Scanning transmission electron microscopy (STEM)

A dual-beam focused ion beam instrument (Helios 5 UX, Thermo Fisher Scientific) was used to prepare a sample of LAO/KTO (111) approximately 30 nm thick. The sample was cut by 30 kV Ga ions and cleaned by 5 kV and 2 kV Ga ions, followed by $Ar^+$ ion milling for 10 min at 500 V in a STEM specimen preparation system (Nanomill 1040, Fischione) prior to STEM imaging. The atom-resolved STEM images were collected by a double spherical aberration-corrected STEM (Spectra Ultra, Thermo Fisher Scientific) operated at 300 kV. The chemical composition was resolved with an energy dispersive X-ray analyzer (Ultra X) in STEM spectrum imaging mode. The STEM–EELS experiments were performed using a Gatan GIF Continuum K3 HR/1069 HR system with an accelerating voltage of 300 kV.

## PFM and SS-PFM

PFM measurements were carried out using a commercial Environmental Atomic Force Microscopy system (Cypher ES, Oxford Instruments) at room temperature in an $N_2$ atmosphere. The PFM and the switching-spectroscopy piezoelectric hysteresis loops were measured in PFM dual-AC resonance tracking mode. A triangular voltage waveform with a frequency of 0.025 Hz was applied to collect both the amplitude and phase signals of modulated tip vibration as a function of the bias voltage, with a maximum of 20 V. The writing of ferroelectric domains was accomplished by scanning the PFM tip over a designated area with a fixed voltage in a nitrogen gas environment to eliminate ion implantation.

## Second harmonic generation

A confocal microscope (WITec, Alpha300RAS) equipped with 1064 nm laser excitation (NPI Rainbow1064 OEM) was employed for non-linear optics measurements (Fig. 2a). The incident light passes through a linear polarizer and a half-wavelength ($\lambda/2$) plate before striking the sample normally. The generated second harmonic signal with a 532 nm wavelength passes through the $\lambda/2$ plate and then the second linear polarizer (analyzer). The $\lambda/2$ plate is rotated in 10° steps for studies of the angular dependence of SHG. For low temperature SHG measurements, the sample is mounted onto a sample holder cooled by an open-cycle liquid nitrogen cryostat capable of reaching temperatures as low as 200 K.

Suppose the polarization of the incident light makes the angle $\phi$ with respect to the $KTaO_3$ $[1\bar{1}0]$ direction, i.e., the x-axis, then the electric field is given by

$$\begin{bmatrix} E_x \\ E_y \\ E_z \end{bmatrix} = \begin{bmatrix} E_0\cos\phi \\ E_0\sin\phi \\ 0 \end{bmatrix} \quad (1)$$

The SHG $d$ matrix for the $m$ point group with electric polarization in $(1\bar{1}0)$ plane is

$$d_m \begin{pmatrix} 0 & 0 & 0 & 0 & d_{15} & d_{16} \\ d_{21} & d_{22} & d_{23} & d_{24} & 0 & 0 \\ d_{31} & d_{32} & d_{33} & d_{34} & 0 & 0 \end{pmatrix} \quad (2)$$

Hence, the SHG signal is given by

$$P^{2\omega} = d_m \begin{pmatrix} E_0^2\cos^2\phi \\ E_0^2\sin^2\phi \\ 0 \\ 0 \\ 0 \\ 2E_0^2\cos^2\phi\,\sin\phi \end{pmatrix} = E_0^2 \begin{pmatrix} -2d_{16}\sin\phi\,\cos\phi \\ d_{21}\cos^2\phi + d_{22}\sin^2\phi \\ d_{31}\cos^2\phi + d_{32}\sin^2\phi \end{pmatrix} \quad (3)$$

Since the SHG light goes through a $\lambda/2$ wave plate before it reaches the analyzer, the SHG light intensities corresponding to x- and y-polarization of the analyzer become

$$I_X^{2\omega} \propto E_0^4 \left( d_{21}\sin\phi\cos^2\phi + d_{22}\sin^3\phi - 2d_{16}\,\sin\phi\cos^2\phi \right)^2 \quad (4)$$

$$I_X^{2\omega} \propto E_0^4 \left( d_{21}\cos^3\phi + d_{22}\sin^2\phi\,\cos\phi + 2d_{16}\sin^2\phi\,\cos\phi \right)^2 \quad (5)$$

The above expression fits nicely with the SHG data shown in Fig. 2.

## Electrical transport measurements

Longitudinal resistance and Hall effect measurements were conducted using the Hall bar pattern shown in Fig. 4a. The width of the Hall bar is 30 μm, and the distance between the two contacts for longitudinal resistance measurement is 100 μm. The excitation current is set at 100 nA. All electric measurements were performed using a physical properties measurement system (PPMS DynaCool 12 T, Quantum Design) with variable sample temperatures ranging from 1.8 to 300 K. For temperatures below 1.8 K, a dilution refrigerator attachment (Quantum Design) is used to reach temperatures as low as 100 mK.

## Data availability

The data that support the findings of this study and all other relevant data are provided with this paper and are also available via the Figshare repository at https://doi.org/10.6084/m9.figshare.30889256

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

## Acknowledgements

This work was supported by National Key R&D Program of China (No. 2023YFA1406400 and No. 2024YFA1408102 to J.W.), Research Center for Industries of the Future (RCIF project No. WU2023C001 to J.W.) at Westlake University, the National Natural Science Foundation of China (Grant No. 12174318 to J.W.), the Zhejiang Provincial Natural Science Foundation of China (Grant No. XHD23A2002 to J.W.). We acknowledge the assistance provided by Dr. Qike Jiang of the Instrumentation and Service Center for Physical Sciences and the Instrumentation and Service Centers for Molecular Science at Westlake University.

## Author contributions

The film synthesis and lithography were done by M.Z., STEM, SHG, PFM and electrical transport measurements were done by M.D.D. and X.B.C., and the analysis and interpretation put forward by J.W.

## Competing interests

The authors declare no competing interests.
