## [Transparent Peer Review file · Nature Communications]

Strongly coupled interface ferroelectricity and interface superconductivity in amorphous LaAlO₃/KTaO₃(111)

Corresponding Author: Professor Jie Wu

Version 0:

Reviewer comments:

Reviewer #1

(Remarks to the Author)

The manuscript titled “Strongly coupled interface ferroelectricity and interface superconductivity in LAO/KTO” reports the observation and manipulation of ferroelectric distortion at the interface between LaAlO₃(LAO) and (111)-oriented KTaO₃ (KTO). The ferroelectricity manifests as relative displacements of K and Ta atoms in KTO near the interface. Application of voltage via an AFM tip flips the polarization of ferroelectricity and gives rise to local domains; such domain formation increases the electrical resistance profoundly and suppresses the superconductivity. Such process is partially reversible with sweeps of the tip voltage, thus guarantees an on-off switch of interfacial charge transport and superconductivity.

Some of these results are interesting, pointing towards good promises from the perspective of device applications. However, the manuscript contains many vague arguments and obscure presentations, which inevitably lead to a certain level of confusion. Furthermore, AFM-tip manipulation of the interfacial electronic state between two insulating oxides has been extensively explored [e.g., C. Cen, et al., Nat. Mater. 7, 298 (2008); Y.-Y. Pai, et al., Phys. Rev. Lett. 120, 147001 (2018); A. Nethewala, et al. Nanoscale Horiz. 4, 1194 (2019); M. Yu, et al., Phys. Rev. X 15, 011037 (2025)]. I doubt the present work has sufficient scientific significance and novelty that can promise its publication in Nature Communications.

Before I can provide a final evaluation of this manuscript, a few questions need to be addressed.

(1) Page 4, lines 94-100. The authors wrote that “The K displacement, δ , is along KTO [110] direction and depends sensitively on its distance from the interface. In contrast, the iDPC-STEM image taken from the KTO(11-2) plane (Extended Data Fig. 2), which is perpendicular to the KTO(1-10) plane in Fig. 1, shows no appreciable Ta displacement. This verifies that the Ta displacement is normal to the KTO(11-2) plane and hence it is precisely along KTO [110] direction.” I found these sentences to be very confusing. First, they initially mentioned “K displacement” but then switched to “Ta displacement” --- are the two terms exchangeable? Or they describe different things? Second, since the whatever displacement is along [110], it is apparently normal to the (110) plane instead of (11-2); hence, I cannot comprehend what the authors meant by saying “This evidences that the K-Ta displacement is entirely along KTO [110] direction and thus has no projection in the (11-2) plane” (caption of Extended Data Fig. 2, line 514-515). The [110] vector definitely has a finite projection in the (11-2) plane.

(2) Still the contexts mentioned in (1). Let’s acknowledge that the atom displacements that causes ferroelectric distortion is along [110], but there are three equivalent [110] directions at the (111) interface, according to its threefold rotational symmetry. In essence, there could correspondingly be three types of ferroelectric domains with the polarizations changing by 120 degrees between each two, but the authors then noted “No ferroelectric domain has been found in the as-grown LAO/KTO(111)...” (line 150). To me, this means rotational symmetry breaking as the system spontaneously selects one special (110) direction for the distortion. Is this true? If so, what is the underlying mechanism? I suggest that discussions on this point must be added, since such a tendency of emergence nematicity can be crucial for understanding the unique properties of KTO-based heterointerfaces [e.g., G. Zhang, et al., Nat. Commun, 14, 3046 (2023)].

(3) As shown in Fig. 4h, the resistance decreases remarkably within a time window of 1 h after AFM-tip writing. Therefore, it may keep change during the cooldown process in PPMS. Is this the cause of the “metallic” behavior displayed in Fig. 4f? Have the authors performed measurement of $\rho(T)$ under thermal cycles and testify the data repeatability? Also, what is origin of the resistive upturn below ~ 30 K in the 0 V, 6 V and -21 V curves?

(4) I have further questions about the model proposed by the authors, that the metallic resistivity stems from either a conducting layer beneath the insulating ferroelectric layer, or percolative paths through it. Indeed, the data presented in Figs. 4h and i point towards an unphysically short electron mean free path of ~ 0.002 nm ($\sim 1/200$ of the lattice constant) at 300 K. I think this can only be reconciled with the latter (1D filamentary path) scenario but cannot be understood with the presence of a 2D conducting layer, because a parallel connection of an insulating layer and a conducting layer must have a resistance lower than the conducting layer. The problem is, however, if the 1D picture is correct, then the 2D Hall number in this regime is meaningless.

(5) The authors attributed the suppression of superconductivity in the high resistance state (driven by negative tip voltage) to enhanced disorder. This seems to be inconsistent with the above-mentioned scenarios: if the ferroelectrically flipped layer (thickness ~ 1.9 nm) is indeed insulating, as implied by the extremely high resistance, the superconductivity is certainly killed therein with no excuse. To me, such an insulating state is more likely driven by the charge depletion related to the change of oxygen vacancies, instead of the disorder-induced localization. It was mentioned in the manuscript that the oxygen vacancies vary with the ferroelectric distortion. This possibility ought to be considered to some extent. Besides, a thickness of 1.9 nm for the dead layer sounds lower than the reported thickness of superconducting layers in KTO-based heterointerfaces [Ref. [24]; X. Hua, et al., npj Quantum Mater. 7, 97 (2022)]. Hence, this is insufficient to explain the absence of superconductivity if the underlying metallic layer is “not directly affected by the ferroelectric flipping” as evidenced by the STEM results. In all, the Hall mobility in Fig. 4i for -21 V is too low to be a meaningful parameter, I do not think an argument of disorder effect (in a metallic state) can be made based on it.

Minor issues

- I suggest the authors to avoid using abbreviations (“LAO/KTO”) in the title because no one really understands what they are.
- Figures 1b and c, the red dashed boxes are almost invisible on the dark background. They should be made clearer.
- The raw data of Hall resistance should be displayed somewhere.

Reviewer #2

(Remarks to the Author)

The authors have investigated LaAlO₃/KTaO₃ (111) interface using a set of techniques and claim that superconductivity and ferroelectricity coexist in the system. While the authors have demonstrated superconductivity and ferroelectricity individually at different temperatures, there is no direct experimental evidence to claim they coexist at the same temperature. I have several other technical comments also. Based on these, I am unable to judge whether the manuscript is suitable for publication in Nature Communications. I request the authors to provide a thorough response to my questions before I make a decision about the manuscript.

1. What is the role of water vapor in the growth process? Does it alter local chemistry of the substrate? What characterizations/measurements lead the authors to maintain a partial pressure of water vapor 10^{-7} mbar during the growth? What happens if the growth is performed without water vapor?
2. The authors claim the interface is sharp. While the EELS shows the intermixing of the length scale of 1 nm. Since the LAO layer is amorphous, the title of the paper is somewhat confusing. There is no experimental report that the film is LaAlO₃, most likely it is La_xAl_yO_z. So, the author should bring this aspect appropriately throughout the text including the title.
3. Regarding SHG: In the method section, the authors have written ‘For low temperature SHG measurements, the sample is mounted onto a sample holder cooled by an open-cycle liquid nitrogen cryostat capable of reaching temperatures as low as 200 K.’ In the main text, SHG data is shown down to much lower temperature (till 1.6 K?). How did the authors achieve 1.6 K in the SHG set up? The polarimetry data is missing at 1.6 K. This is essential as it will show the presence of polar distortion (if present) in the superconducting phase, which is the main claim of the paper. Fig. 2d shows that the SHG signal is actually decreasing below 120 K. Any discussion on this aspect is missing in the text.
4. The authors have performed resistance measurement by applying a gate voltage through PFM tips and shown the hysteresis in R vs. V_{tip} . Under a gate voltage, charge trapping can happen due to gate voltage induced band bending, oxygen vacancy migrations etc., which has been demonstrated extensively in the context of SrTiO₃ based heterostructures [Sci. Rep. 4, 6788 (2014); Phys. Rev. Lett. 124, 017702 (2020); Phys. Rev. Applied 15, 054008 (2021)]. Such charge trapping phenomena also give rise to hysteresis in resistance vs. gate voltage measurement. The authors have not considered these aspects at all in the explanation of their observations.
5. Regarding the data presented in Figure 4(e), was the resistance measured using a two-probe or four-probe technique? If a four-probe method was employed, please explain how such high resistance values could be accurately determined.
6. Regarding the data plotted in Figure 1f, was it acquired from a specific localized region? Additionally, please clarify the methodology used to determine the error bars.
7. The introduction of the paper started with the discussion of the polar metal. However, the paper ends with the effect of gate voltage modulation of the carrier density, which controls superconductivity. This has been demonstrated already in earlier literature.

In summary, while the experiments confirm the presence of ferroelectricity at 300 K using SHG, PFM, and STEM, the

manuscript does not directly show the presence of ferroelectricity in the superconducting state.

Reviewer #3

(Remarks to the Author)

The manuscript discusses a putative coexistence of ferroelectricity and superconductivity at the LaAlO₃/KTaO₃(111) interface. The manuscript is clear and well-written, and if the claim is verified, it certainly merits publication in Nature Communications, as it is at the forefront of current literature in the field of oxide heterointerfaces and related applications in electronics and materials science.

However, I believe the claim is not adequately supported by the evidence presented in its current form. I am particularly concerned about the issue of oxygen vacancies (OxVac) at the interface, as noted by the authors too.

While the superconducting state appears indisputable and is also supported by literature (APL Mater. 11, 121108, 2023), the ferroelectric (FE) ordering is much more subtle. It is well-known that FE measurements are elusive and prone to misleading results (J. Phys.: Condens. Matter 20 (2008) 021001), and in my opinion, the three pieces of evidence provided by the authors (STEM, SHG, and PFM) are insufficient to make a definitive claim.

The first strong evidence of FE ordering would be detecting a FE phase-transition critical temperature, which was not explored in this manuscript. Another critical point is the switchability of polarization, claimed in the manuscript through PFM writing. However, regarding the LaAlO₃/SrTiO₃ interface, Bark et al. write (Nano Lett. 2012, 12, 1765–1771):

"The PFM response we have detected in LAO is very similar to that observed in ferroelectric materials. Its most likely origin, however, is a spatial polarization induced in the LAO layer due to the redistribution of oxygen vacancies."

I suspect a similar phenomenon is occurring here. Indeed, other authors (App. Phys. Lett. 103, 062902, 2013) have used the stability of polarization contrast over several hours as proof of the true FE nature of the system:

"However, these results represent a necessary but not sufficient proof in favor of room temperature ferroelectricity in our films. As discussed in Ref. 18, there are two possible alternative explanations: (1) simple charging of the surface and (2) ionic displacements of the oxygen ions perpendicular to the surface in a direction defined by the applied electric field. For this reason, we investigated the time dependence of the amplitude of the piezo-response signal, which in the case of ferroelectric materials is proportional to the polarization."

Here Ref. 18 is the aforementioned Nano Lett. paper.

According to the authors of this manuscript: "It should be mentioned that after the writing process, the effect of tip writing decays over time, quite significantly for $V_{tip} < -10$ V." Moreover, the authors acknowledge the significant role of OxVac in their observations.

The issue is that both PFM and SHG signals are not exclusively related to FE ordering; they can have various other sources. One possible way to investigate this aspect is to anneal the sample in an oxygen-rich atmosphere (even simply in air) to significantly reduce the OxVac density, disentangling intrinsic and extrinsic effects, as reported in Appl. Phys. Lett. 104, 261603 (2014). For a genuine FE ordering claim, the SHG signal should return to (almost) its initial state after the annealing process.

One minor but not negligible issue I see, is that the SHG signal of the interface was compared to a pristine substrate. However, this is not the appropriate comparison since the pristine substrate likely has no, or negligible, OxVac density at the surface. The proper comparison would be between the investigated interface and a substrate subjected to the same pressure, temperature, and time in the PLD chamber, but without the laser pulse to induce epitaxy. Alternatively, the comparison should be made with at least a doped substrate with a comparable OxVac density.

Finally, the issue of FE domain size: SHG and PFM have very different spatial resolutions, with the former being much larger than the latter. It is possible to have domains too small to be seen by SHG but easily detected by PFM. Conversely, if the domains are very large, they could exceed the total scanned area of PFM, but in this case, SHG should detect them. If they are absent in both techniques, it is quite strange. How can a FE material have no spontaneous domain structure? The idea of a single domain state forming spontaneously is quite unlikely, especially in 2D FE, where domain size is known to be of the same order of magnitude as the interface thickness. The PFM writing could be attributed to OxVac accumulation effects. The fact that the tip writing decays over time is a strong indication that the signal results from ion migration rather than FE ordering. Finally, the hysteretic loops could also be due to trapping/untrapping mechanisms at the interface, modulating the interfacial electric field.

In summary, I believe the manuscript cannot be accepted in its present form, nor by merely changing its text. It requires further experimental work to confirm the proposed claim. However, I think the importance of the findings, if confirmed, justifies the effort.

Minor points:

17 - "we discover that ferroelectric order resides at the interface..." -> "we found evidence of a ferroelectric order at the interface..."

22 - Superconductivity cannot "diminish and reappear": the critical temperature can be higher or lower with respect to a certain value, the material can undergo a phase transition so that it is not superconductive, even if only for a short period of time. This sentence is quite misleading in its present form.

25 - "STEM reveals K ions are displaced relative to" -> "STEM reveals the displacement of K ions relative to"

26 - "electric" -> "ferroelectric"

28 - "hysteresis change" -> "hysteretical change"

73 - "postulated to be related" -> "attributed"

- Problem of beam damaging in STEM measures, how do they know when it is the correct time to stop?

- 124: the SHG intensity is proportional to the square of the fundamental intensity (eq16 in Ref36).

135-137: this statement is correct only if one can link the SHG signal to FE order alone. In case the SHG would be found to

be of structural origin (i.e. similar on oxvac samples with no FE order) then the statement is not correct.

Version 1:

Reviewer comments:

Reviewer #1

(Remarks to the Author)

I appreciate that the authors made comprehensive responses to my questions. After reading the updated version of manuscript, I found that the data presentation and corresponding discussions are sufficiently improved; the potential influence of oxygen vacancies has been carefully analyzed via the performance of supplementary experiments, which allows to distinguish the contributions from ferroelectric polarization and oxygen vacancy redistribution. This information is very helpful to the readers for understanding the mechanism of the PFM-tip manipulation process. Nevertheless, as I mentioned in the first round of review, microscale (or even nanoscale) charge writing and ferroelectric programming exploiting an AFM tip have been extensively realized in oxide interface systems. Modulation of superconductivity via creating ferroelastic domains has also been reported [for instance, Y.-Y. Pai, et al., Phys. Rev. Lett. 120, 147001 (2018)]. In the present work, the writing and erasing of non-superconducting region are achieved by the creation and elimination of ferroelectric domain with flipped polarization vector, respectively; such domains are characterized by reduced carrier density and boosted impurity scattering rate, indeed, they are most likely to be insulating owing to profound localization effect. As such, the control of superconductivity is actually engendered by introducing an insulating phase at the interface. This, in my opinion, can hardly be regarded as a significant breakthrough on the basis of existing results. I tend to suggest that this manuscript is suitable for a more specialized journal.

Reviewer #2

(Remarks to the Author)

While the authors have attempted to answer the questions that I asked during the first round of review, two major questions have not been addressed adequately, in my opinion.

Issue 1: Regarding the coexistence of ferroelectricity at the superconducting phase: The authors have argued by showing the power dependence of SGH data at 2 K that the sample has polar distortion. However, the resistance at 2 K is 180 K, which clearly establishes that the sample is metallic, not superconducting. Thus, the author's claim of the coexistence of ferroelectricity and superconductivity at the same temperature is not experimentally demonstrated.

Issue 2: Role of charge trapping/detrapping under electric field: In spite of acknowledging that this is a possible scenario, the authors have not shown any experimental data to rule out this important possibility.

Overall, I do not think that the reported experimental results directly confirm the author's claim, and I do not recommend the publication of this in Nature Communications in its present form.

Reviewer #3

(Remarks to the Author)

The authors have undertaken a thorough and careful revision of their manuscript, incorporating not only substantial improvements to the text but also presenting several new and compelling experimental findings. Figure R7 (SHG vs T) stands out as particularly intriguing, displaying an unusual and noteworthy behavior. I strongly encourage the authors to further explore this phenomenon in future studies. Given the significant effort invested in strengthening their claims, I believe the paper merits publication in this journal. I also hope it will inspire further research into the role of OxVacs in this and related compounds.

Version 2:

Reviewer comments:

Reviewer #2

(Remarks to the Author)

Based on the new experimental data provided and the comments of other reviewers, I recommend the publication of this manuscript. I appreciate the efforts of the authors in responding to the questions.

Reviewer #3

(Remarks to the Author)

In my view, the revised manuscript adequately addresses both my concerns and those of the other reviewers regarding the experimental validation of the paper's claims. The work now appears sound and well-supported in its conclusions. That said, the issue raised by Reviewer 1 concerning the broader or narrower appeal of the topic is, to some extent, subjective. In my opinion, this matter should be decided by the editors rather than the reviewers, given that all three reviewers have

accurately outlined the current state-of-the-art.

Point-by-point response to the referees' comments.

Reviewer #1 (Remarks to the Author):

Comment 1: *The manuscript titled “Strongly coupled interface ferroelectricity and interface superconductivity in LAO/KTO” reports the observation and manipulation of ferroelectric distortion at the interface between LaAlO₃(LAO) and (111)-oriented KTaO₃(KTO). The ferroelectricity manifests as relative displacements of K and Ta atoms in KTO near the interface. Application of voltage via an AFM tip flips the polarization of ferroelectricity and gives rise to local domains; such domain formation increases the electrical resistance profoundly and suppresses the superconductivity. Such process is partially reversible with sweeps of the tip voltage, thus guarantees an on-off switch of interfacial charge transport and superconductivity.*

Some of these results are interesting, pointing towards good promises from the perspective of device applications. However, the manuscript contains many vague arguments and obscure presentations, which inevitably lead to a certain level of confusion. Furthermore, AFM-tip manipulation of the interfacial electronic state between two insulating oxides has been extensively explored [e.g., C. Cen, et al., Nat. Mater. 7, 298 (2008); Y.-Y. Pai, et al., Phys. Rev. Lett. 120, 147001 (2018); A. Nethewwala, et al. Nanoscale Horiz. 4, 1194 (2019); M. Yu, et al., Phys. Rev. X 15, 011037 (2025)]. I doubt the present work has sufficient scientific significance and novelty that can promise its publication in Nature Communications.

Before I can provide a final evaluation of this manuscript, a few questions need to be addressed.

Response 1: The Referee gives a nice summary of our work. While it is true that using an AFM-tip to manipulate interfacial electronic states is not a new idea—and we have never claimed otherwise—the focus of our study lies in interfacial ferroelectricity and ferroelectric superconductivity, which are important for both fundamental research and potential applications. Reported candidates for ferroelectric superconductivity are scarce in the literature, and the discovery of ferroelectric superconductivity at the interface of parent compounds may open new opportunities for exploring novel superconducting systems.

Comment 2: *Page 4, lines 94-100. The authors wrote that “The K displacement, δ , is along KTO [110] direction and depends sensitively on its distance from the interface. In contrast, the iDPC-STEM image taken from the KTO(11-2) plane (Extended Data Fig. 2), which is perpendicular to the KTO(1-10) plane in Fig. 1, shows no appreciable Ta displacement. This verifies that the Ta displacement is normal to the KTO(11-2) plane and hence it is precisely along KTO [110] direction.” I found these sentences to be very confusing. First, they initially mentioned “K displacement” but then switched to “Ta displacement” --- are the two terms exchangeable? Or they describe different things? Second, since the whatever displacement is along [110], it is apparently normal to the (110) plane instead of (11-2); hence, I cannot comprehend what the authors meant by saying “This*

evidences that the K-Ta displacement is entirely along KTO [110] direction and thus has no projection in the (11-2) plane” (caption of Extended Data Fig. 2, line 514-515). The [110] vector definitely has a finite projection in the (11-2) plane.

Response 2: We thank the Referee very much for pointing out these discrepancies, and have revised the text accordingly to avoid confusion. In the original manuscript, the terms “K displacement” and “Ta displacement” were used interchangeably. Although the displacement between K and Ta atoms is, in some sense, relative, we now use “K displacement” consistently throughout the revised manuscript. This change reflects the fact that oxygen atoms in the Ta-O plane show no displacement relative to the Ta atoms (the original Extended Data Fig. 2; now Fig. S2 in the Supplementary Information); strictly speaking, it is the K atoms that are displaced relative to the Ta and O lattice.

The Referee is correct that the [110] vector has a projection onto the (11-2) plane, and we apologize for the earlier inaccuracy. A clear K displacement along [110] is visible in the (-110) plane (Fig. 1). We rotated the viewing angle to check for components orthogonal to [110], examining the (11-2) plane, which is perpendicular to the (-110) plane. No displacement was observed along [-110] (the original Extended Data Fig. 2; now Fig. S2 in the Supplementary Information), confirming that the displacement lies entirely along [110]. We have revised the text accordingly to ensure accuracy.

Comment 3: *Still the contexts mentioned in (1). Let’s acknowledge that the atom displacements that causes ferroelectric distortion is along [110], but there are three equivalent [110] directions at the (111) interface, according to its threefold rotational symmetry. In essence, there could correspondingly be three types of ferroelectric domains with the polarizations changing by 120 degrees between each two, but the authors then noted “No ferroelectric domain has been found in the as-grown LAO/KTO(111)...” (line 150). To me, this means rotational symmetry breaking as the system spontaneously selects one special (110) direction for the distortion. Is this true? If so, what is the underlying mechanism? I suggest that discussions on this point must be added, since such a tendency of emergence nematicity can be crucial for understanding the unique properties of KTO-based heterointerfaces [e.g., G. Zhang, et al., Nat. Commun, 14, 3046 (2023)].*

Response 3: We greatly appreciate the Referee’s insightful comment. In the out-of-plane direction, the electric field \vec{E} , generated by charge transfer across the LAO/KTO interface, breaks the up-down symmetry, making the lowest energy state for the ferroelectric polarization antiparallel to \vec{E} (Fig. R1). The in-plane direction, retains degenerate energy states if the rotational symmetry is preserved. Emergent nematicity provides an intriguing explanation for this puzzle. In addition to the reference cited by the Referee, our group recently reported direct evidence of electronic nematicity in LAO/KTO(111) [Cheng, X. B. et al. Electronic nematicity in interface superconducting LAO/KTO(111). *Phys. Rev. X* **15**, 021018 (2025)]. We used the angle-resolved resistivity method to measure the anisotropy of electrical transport along various in-plane directions and found compelling evidence for electronic nematicity in its normal state, superconducting fluctuating state, and quantum metal state, confirming the absence of ferroelectric domain in pristine

LAO/KTO(111). Using AFM-tip gating, ferroelectric polarization can be locally flipped and domains can be induced (Fig. 3 and 4b). This discussion has been incorporated into the revised manuscript.

Comment 4: *As shown in Fig. 4h, the resistance decreases remarkably within a time window of 1 h after AFM-tip writing. Therefore, it may keep change during the cooldown process in PPMS. Is this the cause of the “metallic” behavior displayed in Fig. 4f? Have the authors performed measurement of $\rho(T)$ under thermal cycles and testify the data repeatability? Also, what is origin of the resistive upturn below ~ 30 K in the 0 V, 6 V and -21 V curves?*

Response 4: Because measuring $R(T)$ from room temperature to 20 mK takes several hours, we waited 10 hours for the effect to stabilize before initiating cooldown. This makes it unlikely that the observed metallic behavior arises from a time-dependent decay of the effect. Repeated measurements of $R(T)$ for the same state yielded overlapping curves. It is likely that the rapid decay of resistance is related to oxygen vacancy redistribution, while the effect of ferroelectric order persists over time.

The resistive upturn below 30 K is presumably due to the localization effect, consistent with the increased disorder scattering in these high-resistance states.

Comment 5: *I have further questions about the model proposed by the authors, that the metallic resistivity stems from either a conducting layer beneath the insulating ferroelectric layer, or percolative paths through it. Indeed, the data presented in Figs. 4h and i point towards an unphysically short electron mean free path of ~ 0.002 nm ($\sim 1/200$ of the lattice constant) at 300 K. I think this can only be reconciled with the latter (1D filamentary path) scenario but cannot be understood with the presence of a 2D conducting layer, because a parallel connection of an insulating layer and a conducting layer must has a resistance lower than the conducting layer. The problem is, however, if the 1D picture is correct, then the 2D Hall number in this regime is meaningless.*

Response 5: What we are suggesting here is that both the reduction of the conducting layer thickness and percolative conductivity contribute to the high resistance state. And the Referee is right that the mean free path for the highest resistance state is shorter than the lattice constant, indicating that percolative conductivity plays a major role. Thus, for the high resistance state, the retrieved Hall number is under influence of percolative connectivity and should be interpreted with caution. We have moved Fig. 4h and 4i to the Supplementary Information and made this point explicit there.

Comment 6: *The authors attributed the suppression of superconductivity in the high resistance state (driven by negative tip voltage) to enhanced disorder. This seems to be inconsistent with the above-mentioned scenarios: if the ferroelectrically flipped layer (thickness ~ 1.9 nm) is indeed insulating, as implied by the extremely high resistance, the superconductivity is certainly killed therein with no excuse. To me, such an insulating state is more likely driven by the charge depletion related to the change of oxygen vacancies, instead of the disorder-*

induced localization. It was mentioned in the manuscript that the oxygen vacancies vary with the ferroelectric distortion. This possibility ought to be considered to some extent. Besides, a thickness of 1.9 nm for the dead layer sounds lower than the reported thickness of superconducting layers in KTO-based heterointerfaces [Ref. [24]; X. Hua, et al., npj Quantum Mater. 7, 97 (2022)]. Hence, this is insufficient to explain the absence of superconductivity if the underlying metallic layer is “not directly affected by the ferroelectric flipping” as evidence by the STEM results. In all, the Hall mobility in Fig. 4i for -21 V is too low to be a meaningful parameter, I do not think an argument of disorder effect (in a metallic state) can be made based on it.

Response 6: We appreciate the Referee’s comments and have improved the discussion on the mechanisms for modulation of conductivity and superconductivity. Incorporating the valuable suggestions of all three Referees, we now propose that the modulation arises from the combined effects of:

1. Flipping of ferroelectric polarization \vec{P} . The work-function imbalance between LAO and KTO drives electron transfer from the LAO film to the KTO substrate, forming a two-dimensional electron gas (2DEG). This interfacial charge transfer creates an internal electric field \vec{E} due to electron depletion on the LAO side and accumulation on the KTO side (Fig. R1). Analogous to a p - n junction, electron accumulation strengthens \vec{E} and suppresses further transfer, with the amount of transferred charge determined by the interfacial chemical-potential balance. Ferroelectric polarization \vec{P} at the LAO/KTO interface introduces an additional field. When its out-of-plane component, \vec{P}_\perp , is antiparallel to \vec{E} , the threshold \vec{E} field for halting transfer increases, yielding higher 2DEG density. When parallel, the 2DEG density decreases. In pristine LAO/KTO(111), \vec{P}_\perp is antiparallel to \vec{E} since this is the lowest energy ground state. Therefore, resistance is low in the pristine state and increases significantly as \vec{P} is flipped.
2. Modification of effective disorder scattering. Beyond carrier-density effects, \vec{P} alters the profile of the potential well confining the 2DEG. Electrons in a narrower potential well experience stronger scattering from disorder, resulting in higher resistance.
3. Redistribution of oxygen vacancies. Ferroelectric polarization correlates with oxygen vacancies at the interface (Fig. 1). The flipping of \vec{P} is inevitably accompanied by the redistribution of oxygen vacancies. Electron trapping by these vacancies renders electron’s mobility and increases resistance.

As this study represents an initial step in exploring ferroelectric superconductivity, it is premature to conclude whether there are additional novel features of ferroelectric superconductivity that may contribute to the modulation process. We therefore focus on our experimental findings while leave open possibilities for future investigation.

Figure R1 | Schematic drawing of the interfacial electric field and ferroelectric polarization \vec{P} . **a**, 2DEG is formed due to charge transfer from LAO to KTO and electrons are confined in a potential well. The interfacial charge transfer creates an internal electric field \vec{E} due to electron depletion on the LAO side and accumulation on the KTO side. **b**, The out-of-plane component \vec{P}_{\perp} , is antiparallel to \vec{E} , for minimization of electrostatic energy. \vec{P}_{\perp} increase the threshold \vec{E} field for halting charge transfer, yielding higher 2DEG density.

Comment 7: Minor issues

- I suggest the authors to avoid using abbreviations (“LAO/KTO”) in the title because no one really understand what they are.

- Figures 1b and c, the red dashed boxes are almost invisible on the dark background. They should be made clearer.

- The raw data of Hall resistance should be displayed somewhere.

Response 7: We’ve revised our manuscript accordingly by changing “LAO/KTO” to “amorphous LaAlO₃/KTaO₃(111)” and enhancing the visibility of the yellow boxes. The raw Hall resistance data has been added to the Supplementary Information.

Reviewer #2 (Remarks to the Author):

Comment 8: The authors have investigated LaAlO₃/KTaO₃ (111) interface using a set of techniques and claim that superconductivity and ferroelectricity coexist in the system. While the authors have demonstrated superconductivity and ferroelectricity individually at different temperature, there is no direct experimental evidence to claim they coexist at same temperature. I have several other technical comments also. Based on these, I am unable to judge whether the manuscript is suitable for publication in Nature Communications. I request the authors to provide a thorough response to my questions before I make a decision about the manuscript.

Response 8: We have added supplementary experimental results and revised discussions to address the Referee’s comments and concerns, with the aim of meeting the Referee’s high standards. The coexistence of superconductivity and ferroelectricity is discussed in Response 11.

Comment 9: What is the role of water vapor in the growth process? Does it alter

local chemistry of the substrate? What characterizations/measurements lead the authors to maintain a partial pressure of water vapor 10^{-7} mbar during the growth? What happens if the growth is performed without water vapor?

Response 9: During sample growth, the water vapor partial pressure (1×10^{-7} mbar) is two orders of magnitude lower than the oxygen partial pressure (1×10^{-5} mbar). At this level, hydrogen incorporation into the KTO(111) substrate is negligible and below the detection limits of techniques such as SIMS.

Atomic force microscopy (AFM) and X-ray diffraction (XRD) analyses confirm that trace water vapor does not produce detectable changes in surface morphology or crystalline structure (Figs. R2a-c). Nevertheless, it enhances the superconducting transition temperature, as shown by $R(T)$ curves comparing samples grown with and without trace water vapor (Fig. R2d). Meanwhile, we also confirmed that a bare KTO(111) substrate exposed to identical water vapor and oxygen levels, and subjected to the same thermal processes without actual LAO film deposition, remained highly insulating. Thus, the trace water vapor likely acts as a catalyst, modifying local surface chemistry to promote better interface formation.

It should also be noted that tiny residual water vapor is commonly present in gas pipelines of deposition systems (e.g., PLD, sputtering) [*P. Scheiderer et al. Phys. Rev. B* **92**, 195422 (2015)], and is known to influence oxide film growth, with controlled introduction improving film quality [*S. Ishizuka et al. J. Appl. Phys.* **100**, 096106 (2006); *S. Ishizuka et al. Jpn. J. Appl. Phys.* **44**, L679 (2005); *T. Li et al. ACS Appl. Mater. Interfaces* **16**, 31237-31246 (2024); *N. Oka et al. Appl. Phys. Express* **5**, 075802 (2012)]. Wet oxidation methods are also routinely used in semiconductor processing to enhance oxide film properties.

Therefore, incorporating $\sim 1\%$ partial pressure of water vapor (1×10^{-7} mbar) during growth improves superconducting performance ($T_c \sim 2.1$ K) [*Z. Chen et al. Science* **372**, 721–724 (2021)]. While further investigation into the surface chemistry evolution would be valuable to elucidate the underlying mechanisms, such work lies beyond the scope of this study.

Figure R2 | Structural and transport properties of LAO/KTO(111) samples grown with and without water vapor. **a** and **b**, AFM images showing the surface morphology of 10 nm LAO/KTaO(111) samples grown without and with water vapor, respectively. **c**, Corresponding XRD spectra are similar for both cases. **d**, Sheet resistance $R_s(T)$ measurements indicate enhanced superconductivity when water vapor was introduced into the growth chamber. The inset amplifies $R_s(T)$ for temperatures below 5 K.

Comment 10: *The authors claim the interface is sharp. While the EELS shows the intermixing of the length scale of 1 nm. Since LAO layer is amorphous, the title of the paper is somewhat confusing. There is no experimental report that the film is LaAlO₃, most likely it is La_xAl_yO_z. So, the author should bring this aspect appropriately through out the text including the title.*

Response 10: We followed the convention established in previous publications, e.g., the first report of KTO-based interface superconductivity (Ref. 22) used the notation LAO/KTO(111) (Ref. 22). We agree that this notation might cause confusion and have therefore changed the title to “a-LaAlO₃/KTaO₃(111)”, where “a” denotes amorphous. For any film deposition method, interfacial atom interdiffusion is inevitable due to entropy. To avoid controversy, the claim of a sharp interface has been removed.

Regarding composition, the PLD target is polycrystalline LaAlO₃ bulk. Although the deposited film’s composition may deviate from that of the target, it is standard practice in the PLD community to refer to the nominal composition. In keeping with this convention, and with previous work on this subject, we denote the film as “amorphous LaAlO₃”. For simplicity and readability, we use the abbreviation “LAO”

for “amorphous LaAlO₃”, which is explicitly stated in the revised manuscript. We have added a paragraph to clarify these points on structure and composition.

Comment 11: *Regarding SHG: In the method section, the authors have written ‘For low temperature SHG measurements, the sample is mounted onto a sample holder cooled by an open-cycle liquid nitrogen cryostat capable of reaching temperatures as low as 200 K.’ In the main text, SHG data is shown down much lower temperature (till 1.6 K?). How did the authors achieve 1.6K in SHG set up? The polarimetry data is missing at 1.6 K. This is essential as it will show the presence of polar distortion (if present) in the superconducting phase, which is the main claim of the paper. Fig. 2d shows that the SHG signal is actually decreasing below 120 K. Any discussion on this aspect is missing in the text.*

Response 11: We employed two SHG instruments installed in two separate cryogenic systems (Fig. R3). One is installed in a WITec, Alpha300RAS cryogenic system (Fig. R3a), which operates down to 77K and features a rotatable polarizer for measuring $I_x^{2\omega}(\phi)$ and $I_y^{2\omega}(\phi)$. This served as our primary SHG instrument. The other is installed in a HORIBA LabRam Odyssey and Attocube attoDRY2100 system (Fig. R3b) that is currently still under development and currently has limited functionality. It can only record SHG signals with the polarizer fixed at an angle, but the HORIBA system can reach temperatures as low as 2 K. SHG data in Fig. 2d of the original manuscript were obtained using the HORIBA setup, while all other SHG data came from the WITec setup.

Although SHG or PFM measurements in a dilution refrigerator would be ideal, we lack access to such facilities and are unaware of commercial systems that offer them. Nevertheless, by combining results from both SHG setups, we obtained compelling evidence for the coexistence of superconductivity and ferroelectricity.

1. I_{SHG} is substantial at 2 K and scales with P^2 (Fig. R3d and R3e), confirming that ferroelectricity is present at 2 K.
2. Superconductivity onsets at 3.5 K (Fig. R3c).
3. $I_{\text{SHG}}(T)$ shows no sign of transition during cooldown (Fig. 2d), indicating that ferroelectricity persists to low temperatures.
4. The orientation of ferroelectric order, determined from $I_x^{2\omega}(\phi)$ and $I_y^{2\omega}(\phi)$, is consistent with TEM and PFM results.
5. Ferroelectric order shall become stronger as thermal fluctuations become weaker at lower temperatures. The possibility of a ferroelectric-to-paraelectric phase transition is further ruled out by the $R(T)$ measurement (the pristine curve in Fig. 4f), which shows no kink indicative of such a transition.

These results and the accompanying discussion have been added to the revised manuscript, further reinforcing our conclusions.

Figure R3 | Two SHG instruments in separate cryogenic systems. **a**, SHG optics installed in the WITec's cryogenic system, with temperature limited to 77 K. **b**, SHG optics installed in the HORIBA LabRam Odyssey and Attocube attoDRY2100 cryogenic system, capable of reaching ~ 2 K. **c**, $R(T)$ of LAO/KTO(111), showing the onset of superconductivity ~ 3.5 K. **d**, I_{SHG} at 2 K as a function of P , the laser power at 1064 nm wavelength. **e**, Quadratic dependence of I_{SHG} on P at 2 K.

Comment 12: The authors have performed resistance measurement by applying a gate voltage through PFM tips and shown the hysteresis in R vs. V_{tip} . Under a gate voltage, charge trapping can happen due to gate voltage induced band bending, oxygen vacancy migrations etc., which had been demonstrated extensively in context of $SrTiO_3$ based heterostructures [Sci. Rep. **4**, 6788 (2014); Phys. Rev. Lett. **124**, 017702 (2020); Phys. Rev. Applied **15**, 054008 (2021)]. Such charge trapping phenomena also gives rise to hysteresis in resistance vs. gate voltage measurement. The authors have not considered this aspects at all in the explanation of their observations.

Response 12: We thank the Referee for highlighting the possible role of charge trapping effect in the observed resistance hysteresis. In LAO/STO, such effect manifests as hysteresis in resistivity and carrier density when the backside gating voltage is ramped. A similar mechanism could occur in LAO/KTO and contribute to the resistance hysteresis observed during tip gating. Motivated by the Referee's suggestion, we plan to conduct systematic studies of charge trapping, an intriguing topic but beyond the scope of the present work.

Regarding the potential role of oxygen vacancies, we have carried out additional experiments and analyses, as detailed in Response 19. We have added these useful references and revised the manuscript to incorporate this discussion.

Comment 13: *Regarding the data presented in Figure 4(e), was the resistance measured using a two-probe or four-probe technique? If a four-probe method was employed, please explain how such high resistance values could be accurately determined.*

Response 13: All resistance measurements were performed using the four-probe method for better accuracy.

A Keithley 2450 sourcemeter supplied a *dc* current and simultaneously measured *dc* voltages while *in-situ* flipping the ferroelectric polarization with the PFM tip. The square-wave method, alternating the current polarity and measuring the corresponding voltage difference, was employed to eliminate contact-voltage offsets. For low resistance samples ($\leq 10^5 \Omega$), the excitation current was 300 nA; for higher resistance, a lower current ~ 10 nA was used. Although the noise is larger with the lower current, the resistance measurement remained reliable with longer averaging times.

Comment 14: *Regarding the data plotted in Figure 1f, was it acquired from a specific localized region? Additionally, please clarify the methodology used to determine the error bars.*

Response 14: The iDPC-STEM imaging at all examined locations yielded consistent results, including the relative displacement between Ta and K atoms, and the presence of oxygen vacancies. The data in Fig. 1f are representative and generic to LAO/KTO(111).

In the iDPC-STEM images, one pixel corresponds to a physical length of 0.0085 nm. For clearly resolved atomic columns, the error bar was estimated as 2×0.0085 nm; for columns with tails, the estimate was 4×0.0085 nm.

Comment 15: *The introduction of the paper started with the discussion of the polar metal. However, the paper ends with the effect of gate voltage modulation of the carrier density, which controls superconductivity. This has been demonstrated already in earlier literature.*

Response 15: The focus of this work is to report the discovery of interfacial ferroelectricity and ferroelectric superconductivity in LAO/KTO(111) heterointerface. The modulation of superconductivity via tip gating demonstrates that interfacial ferroelectricity can be switched by an external electric field—an effect still elusive in most of the polar metals (hence the term “polar” rather than “ferroelectric” metal, Ref. 8). This also confirms that ferroelectricity and superconductivity are coupled, enabling non-violate control of superconductivity through ferroelectricity.

Our study represents an initial step in exploring ferroelectric superconductivity—a

novel state with both fundamental significance and potential for multifunctional electronics. This work easily extends to topics such as the pairing symmetry of ferroelectric superconductors, where inversion-symmetry breaking rules out conventional *s*-, *p*-, and *d*-wave pairing and allows for unconventional pairings. Given the limited current understanding, ferroelectric superconductivity may host entirely new phenomena, and we hope this work lays the foundations for future advances.

Comment 16: *In summary, while the experiments confirm the presence of ferroelectricity at 300 K using SHG, PFM, and STEM, the manuscript does not directly show the presence of ferroelectricity in the superconducting state.*

Response 16: We conducted comprehensive *in situ* experiments combining resistance measurements and SHG characterization at temperatures down to 2 K. At 2 K, SHG signals from LAO/KTO(111) were detected concurrently with its superconducting transition, as detailed in Response 11. These results demonstrate that ferroelectricity coexists with superconductivity. We trust that these extensive efforts address the Referee's concerns.

Reviewer #3 (Remarks to the Author):

Comment 17: *The manuscript discusses a putative coexistence of ferroelectricity and superconductivity at the LaAlO₃/KTaO₃(111) interface. The manuscript is clear and well-written, and if the claim is verified, it certainly merits publication in Nature Communications, as it is at the forefront of current literature in the field of oxide heterointerfaces and related applications in electronics and materials science.*

Response 17: We thank the referee very much for recognition of the value and impact of our work.

Comment 18: *However, I believe the claim is not adequately supported by the evidence presented in its current form. I am particularly concerned about the issue of oxygen vacancies (OxVac) at the interface, as noted by the authors too. While the superconducting state appears indisputable and is also supported by literature (APL Mater. 11, 121108, 2023), the ferroelectric (FE) ordering is much more subtle. It is well-known that FE measurements are elusive and prone to misleading results (J. Phys.: Condens. Matter 20 (2008) 021001), and in my opinion, the three pieces of evidence provided by the authors (STEM, SHG, and PFM) are insufficient to make a definitive claim.*

Response 18: We agree with the Referee that caution is warranted when claiming ferroelectricity, and we appreciated the amusing reference provided—though we are sure that a banana could not produce the STEM, SHG, and PFM signals we've observed!

Following the Referee's suggestions, we conducted supplementary experiments to examine alternative explanations of the results. Considering both the original and new data, we are confident that switchable ferroelectric order exists at the

LAO/KTO heterointerface. A brief summary of the supporting experimental evidence is provided in Response 23.

Comment 19: *The first strong evidence of FE ordering would be detecting a FE phase-transition critical temperature, which was not explored in this manuscript. Another critical point is the switchability of polarization, claimed in the manuscript through PFM writing. However, regarding the LaAlO₃/SrTiO₃ interface, Bark et al. write (Nano Lett. 2012, 12, 1765–1771):*

"The PFM response we have detected in LAO is very similar to that observed in ferroelectric materials. Its most likely origin, however, is a spatial polarization induced in the LAO layer due to the redistribution of oxygen vacancies."

I suspect a similar phenomenon is occurring here. Indeed, other authors (App. Phys. Lett. 103, 062902, 2013) have used the stability of polarization contrast over several hours as proof of the true FE nature of the system:

"However, these results represent a necessary but not sufficient proof in favor of room temperature ferroelectricity in our films. As discussed in Ref. 18, there are two possible alternative explanations: (1) simple charging of the surface and (2) ionic displacements of the oxygen ions perpendicular to the surface in a direction defined by the applied electric field. For this reason, we investigated the time dependence of the amplitude of the piezo-response signal, which in the case of ferroelectric materials is proportional to the polarization."

Here Ref. 18 is the aforementioned Nano Lett. paper.

Response 19: We thank the Referee for noting that the PFM writing might arise from the redistribution of oxygen vacancies and for suggesting a practical method to distinguish different mechanisms. Following these valuable suggestions, we performed supplementary measurements of the time decay of domain contrast after writing.

Figure R4 | Time decay of the PFM amplitude. A concentric square pattern was written onto LAO/KTO(111) sample at time zero, and the PFM amplitude at the center square was monitored over time. A substantial PFM signal persists beyond 24 hours, indicating a stable ferroelectric polarization. The corresponding PFM and EFM images after 10, 720, and 1440 minutes are shown in the upper and lower inset panels, respectively. While the PFM contrast remains with time, the EFM contrast diminishes by 720 minutes, supporting the interpretation that the EFM signal arises from surface charge or oxygen vacancy effects, whereas the PFM signal originates from ferroelectricity.

As shown in Fig. R4, after writing concentric square patterns with +15 and -15 V, the PFM amplitude decreases rapidly from 2.7 to 1.5 in the first 1 hour, then more gradually from 1.5 to 0.5 between 1 and 12 hours, after which it stabilized. PFM images taken at 10, 720, and 1440 minutes (upper inset) corroborate these trends. In stark contrast, EFM images acquired at the same time intervals (lower inset) show rapid contrast decay, vanishing by 720 minutes (note the identical coloring of the central square and its surroundings at 720 and 1440 minutes). From these observations and the reference cited by the Referee, we infer:

1. The persistent PFM signal originates from ferroelectric polarization, reversible via tip gating.
2. The decaying components of the PFM and EFM signals likely relate to oxygen-vacancy redistribution.

More compelling evidence comes from time-decay measurements of SHG signals after writing. The SHG images (Fig. R5) display clear contrast between regions written with +15 and -15 V applied to the PFM tip. With both the incident laser and

the generated second harmonic light normal to the film surface, and their polarizations in-plane (Fig. 2a), the SHG signal is sensitive only to the in-plane component of electric polarization. Since charge transfer between LAO and KTO produces an out-of-plane electric polarization, the observed SHG contrast mainly reflects the in-plane component of ferroelectric order along [110] (Fig. 1). As the out-of-plane component was flipped using the tip gating, the in-plane component was coupled with the out-of-plane component and was flipped concurrently, resulting in modified contrast in SHG images (Fig. R5a). Importantly, the SHG contrast persists after 70 hours (Fig. R5b), consistent with the PFM results in Fig. R4. Together, these findings provide unambiguous evidence that LAO/KTO(111) possesses a switchable ferroelectric polarization.

Figure R5 | SHG images after pattern writing with the PFM tip. Concentric squares were written by applying +15 V and -15 V voltages on the PFM tip. SHG images were acquired within 4 hours (a) and after 70 hours (b) of writing. The persistent contrast after 70 hours indicates a stable, switchable ferroelectric order in LAO/KTO(111).

To strengthen our case, we performed a comparative experiment on LaAlO₃/MgO(001) to verify that the LAO film itself, or its interface with another oxide, would not trivially produce the PFM and SGH signals observed in LAO/KTO(111).

The LAO/MgO(001) film was grown under identical conditions to the superconducting LAO/KTO(111) samples. Applying ± 30 V on the PFM tip—well above the voltage applied for LAO/KTO(111)—we wrote a concentric square pattern on LAO/MgO.

Figure R6 | PFM (a), EFM (b), and SHG (c) images of the LAO/MgO(001) sample after writing a concentric square pattern with the PFM tip. No SHG signal was detected, and the EFM contrast decayed within 48 hours (d), in sharp contrast to that observed on LAO/KTO(111).

Comparison of the results for LAO/KTO(111) (Figs. R4 and R5) and LAO/MgO(001) (Fig. R6) leads to the following conclusions:

1. PFM and EFM contrast appear in LAO/MgO but vanish within 48 hours, indicating that these signals likely arise from surface charging or oxygen-vacancy effects. The EFM signal of LAO/KTO also decays over time, presumably for the same reason. However, the time-stable component of the LAO/KTO PFM signal points to the presence of switchable ferroelectric polarization.
2. No SHG signal was detected from LAO/MgO, even immediately after pattern writing, in sharp contrast to the stable SHG signal from LAO/KTO. This indicates that surface charging or oxygen vacancies do NOT generate SHG signals, and that the SHG signal from LAO/KTO originates from ferroelectric polarization.

Comment 20: According to the authors of this manuscript: "It should be mentioned that after the writing process, the effect of tip writing decays over time, quite significantly for $V_{tip} < -10$ V." Moreover, the authors acknowledge the significant role of OxVac in their observations.

The issue is that both PFM and SHG signals are not exclusively related to FE ordering; they can have various other sources. One possible way to investigate this aspect is to anneal the sample in an oxygen-rich atmosphere (even simply in air) to significantly reduce the OxVac density, disentangling intrinsic and extrinsic effects, as reported in *Appl. Phys. Lett.* 104, 261603 (2014). For a genuine FE ordering claim, the SHG signal should return to (almost) its initial state after the

annealing process.

Response 20: We followed the Referee's suggestion and annealed the samples in air. While ramping the temperature from room temperature to 450°C, I_{SHG} manifests a pronounced peak around 300°C for both LAO/KTO(111) samples (Fig. R7), indicative of a change in the structure or chemical stoichiometry. This behavior complicates the data analysis and warrants further investigation. Nevertheless, the SHG signal survives the annealing process, albeit with reduced intensity, and the SHG symmetry remains unchanged upon returning to room temperature.

We are not surprised that the ferroelectric order varies with the concentration of oxygen vacancies. As noted in the original manuscript, ferroelectricity at the LAO/KTO interface must be correlated with oxygen vacancies at the interface (Fig. 1). It is also well established in the literature that oxygen vacancies can enhance ferroelectric polarization or piezoelectricity in perovskite oxide films (e.g., see *Sci. Adv.* **8**, eabm8550 (2022) and *Science* **375**, 653 (2022)). Thus, it is consistent with expectations that the strength of ferroelectric polarization changes with oxygen vacancy density. However, the key point here is that the observed SHG signal originates from ferroelectric order, rather than directly from oxygen vacancies.

Unlike the experimental setup used in the reference paper (*Appl. Phys. Lett.* **104**, 261603 (2014)), in our SHG optical configuration both the incident laser and the generated second-harmonic light are normal to the film surface, and the polarizations of both are in-plane (Fig. 2a). As a result, our SHG setup is sensitive only to the in-plane component of electric polarization. The charge transfer between LAO and KTO, and the resultant electric polarization, are orientated along the out-of-plane direction. Therefore, the SHG signal in our measurements is mainly contributed by the in-plane component of ferroelectric polarization along the [110] direction (Fig. 1). Furthermore, we have already demonstrated in Fig. R6 that oxygen vacancies in LAO/MgO(001) generate NO detectable SHG signal. Therefore, the SHG signal of LAO/KTO(111) can be attributed to ferroelectric order.

Figure R7 | SHG intensity of LAO/KTO(111) during air annealing for the sample #1 (a) and #2 (b). I_{SHG} manifests a pronounced peak during heating, indicative of a change in the structure or chemical stoichiometry.

Comment 21: One minor but not negligible issue I see, is that the SHG signal of the interface was compared to a pristine substrate. However, this is not the appropriate comparison since the pristine substrate likely has no, or negligible,

OxVac density at the surface. The proper comparison would be between the investigated interface and a substrate subjected to the same pressure, temperature, and time in the PLD chamber, but without the laser pulse to induce epitaxy. Alternatively, the comparison should be made with at least a doped substrate with a comparable OxVac density.

Response 21: Our aim in comparing the SHG signals from the LAO/KTO(111) sample and the KTO(111) substrate is to demonstrate that the KTO(111) substrate itself does not generate SHG signal (Fig. R8), and the measured SHG signal of LAO/KTO(111) therefore originates from the heterointerface. This comparison is necessary because the inversion symmetry of KTO(111) is broken at its surface, which, in principle, could allow SHG to emerge.

Following the Referee's suggestion, and we carried out the corresponding experiment. The KTO(111) substrate was heated to 300°C under conditions identical to those used for LAO film deposition. The annealed KTO(111) substrate produced only a weak SHG signal (Fig. R8). We also attempted to write patterns with tip voltage ramped all the way up to ± 50 V, but observed no contrast in either PFM or EFM images after writing, indicating that the ferroelectric polarization was too weak to be detected by PFM or EFM.

As proposed in the manuscript, oxygen vacancies at the LAO/KTO interface presumably affect the relative rotation of the oxygen octahedra, as well as the Ta-O-Ta bond length and angle, thereby inducing interfacial ferroelectricity. For the annealed KTO(111) substrate, the same mechanism may also operate: oxygen vacancies generated at the surface during annealing give rise to a weak ferroelectric polarization. This is consistent with literature reports that oxygen vacancies can enhance ferroelectric polarization or piezoelectricity in perovskite oxide films (e.g., see Sci. Adv. 8, eabm8550 (2022) and Science 375, 653 (2022)).

Figure R8 | SHG signals for the KTO(111) substrate, annealed KTO(111) substrate, and LAO/KTO(111) sample. The wavelength of the laser is 1064 nm and the SHG light shall have a wavelength of 532 nm.

Comment 22: Finally, the issue of FE domain size: SHG and PFM have very different spatial resolutions, with the former being much larger than the latter. It is possible to have domains too small to be seen by SHG but easily detected by PFM. Conversely, if the domains are very large, they could exceed the total scanned area of PFM, but in this case, SHG should detect them. If they are absent in both

techniques, it is quite strange. How can a FE material have no spontaneous domain structure? The idea of a single domain state forming spontaneously is quite unlikely, especially in 2D FE, where domain size is known to be of the same order of magnitude as the interface thickness. The PFM writing could be attributed to OxVac accumulation effects. The fact that the tip writing decays over time is a strong indication that the signal results from ion migration rather than FE ordering. Finally, the hysteretic loops could also be due to trapping/untrapping mechanisms at the interface, modulating the interfacial electric field.

Response 22: As we've demonstrated, ferroelectric domains created by PFM tip writing can be readily resolved by both PFM and SHG methods. Thus, the absence of ferroelectric domains is not due to any malfunction of the techniques employed. If the ferroelectric domain size is close to the interface thickness as the Referee suggested, then it would be about 1.9 nm, which is the thickness of the interfacial ferroelectric phase determined by the iDPC-STEM measurements (Fig. 1). Given the spatial resolution of PFM (~25 nm) and SHG microscopy (~1.5 μm), it's unsurprising that domains of this size cannot be resolved by either method. However, from the images taken after pattern writing (Fig. 3), the unpatterned outside region appears to have the same contrast as the inner square patterned with +10 V tip voltage. This likely implies that the pristine state is indeed a single domain.

We are therefore inclined towards the interpretation proposed by the Referee 1 that an emergent electronic nematicity spontaneously breaks the in-plane rotational symmetry, rendering the three [110] directions at the (111) interface inequivalent. This provides a preferred direction for the ferroelectric order. Indeed, we have previously discovered nematic superconductivity and nematic normal state in LAO/KTO(111) [Cheng, X. B. *et al.* Electronic nematicity in interface superconducting LAO/KTO(111). *Phys. Rev. X* **15**, 021018 (2025)]. Please refer to Comment 3 and Response 3 for detailed discussion.

For discussions of oxygen vacancies and time decay, see Responses 18,19 and 20.

Comment 23: *In summary, I believe the manuscript cannot be accepted in its present form, nor by merely changing its text. It requires further experimental work to confirm the proposed claim. However, I think the importance of the findings, if confirmed, justifies the effort.*

Response 23: Here, we briefly summarize the key experimental evidence supporting a switchable ferroelectric order in LAO/KTO(111):

1. STEM images reveal the K atom displacement relative to the Ta-O lattice (Fig. R9). This directly demonstrates that the centers of positive and negative ions are displaced relative to each other—the defining characteristic of ferroelectricity.
2. STEM images also show that ferroelectric order is coupled to oxygen vacancies at the interface. Consequently, flipping the ferroelectric polarization also redistributes oxygen vacancies, likely responsible for the rapid decaying component of the PFM amplitude and resistance over time.
3. A substantial portion of the PFM amplitude and resistance change after

- flipping is stable against time decay, indicating that it originates from ferroelectric polarization.
- Due to the configuration of our SHG optics, the SHG signal is sensitive only to the in-plane component of ferroelectric polarization. Thus, the measured SHG signals and images directly reflect the ferroelectric order.
 - Taken together STEM, PFM and SHG results confirm that ferroelectric polarization emerges at the LAO/KTO(111) heterointerface and it can be flipped by poling.

We have also revised our discussions on the mechanisms underlying modulations of conductivity and superconductivity to explicitly incorporate contributions from ferroelectric order, oxygen vacancies, and disorder. We are grateful for the Referee's suggestions, and we hope that with these improvements, which required substantial efforts, this work now meets the Referee's high standards.

Figure R9 | Projections of polar KTO lattice in the (1-10) (a), (11-2) (b), and (111) (c) planes according to STEM images. The purple solid line represents the angle-dependent SHG intensity $I_x^{2\omega}(\phi)$ at room temperature.

Comment 24: Minor points:

17 - "we discover that ferroelectric order resides at the interface..." -> "we found evidence of a ferroelectric order at the interface..."

22 - Superconductivity cannot "diminish and reappear": the critical temperature can be higher or lower with respect to a certain value, the material can undergo a phase transition so that it is not superconductive, even if only for a short period of time. This sentence is quite misleading in its present form.

25 - "STEM reveals K ions are displaced relative to" -> "STEM reveals the displacement of K ions relative to"

26 - "electric" -> "ferroelectric"

28 - "hysteresis change" -> "hysteretical change"

73 - "postulated to be related" -> "attributed"

- Problem of beam damaging in STEM measures, how do they know when it is the correct time to stop?

- 124: the SHG intensity is proportional to the square of the fundamental intensity (eq16 in Ref36).

135-137: this statement is correct only if one che link the SHG signal to FE order alone. In case the SHG would be found to be of structural origin (i.e. similar on oxvac samples with no FE order) then the statement is not correct.

Response 24: We thank the Referee for pointing out these issues and we have

carefully incorporated all the recommended linguistic improvements in the revised text.

Regarding beam damage, it is a trial-and-error process. We monitored the STEM image during imaging until we saw the damage occurred. Then, next time we controlled the electron dose to be below the threshold value.

Regarding SHG signal, please refer to Response 11, 19, 20 and 21 for detailed discussions. We have included these experimental results and discussions in the revised manuscript and Supplementary Information.

Point-by-point response to the referees' comments.

Reviewer #1 (Remarks to the Author):

Comment 1: I appreciate that the authors made comprehensive responses to my questions. After reading the updated version of manuscript, I found that the data presentation and corresponding discussions are sufficiently improved; the potential influence of oxygen vacancies has been carefully analyzed via the performance of supplementary experiments, which allows to distinguish the contributions from ferroelectric polarization and oxygen vacancy redistribution. This information is very helpful to the readers for understanding the mechanism of the PFM-tip manipulation process. Nevertheless, as I mentioned in the first round of review, microscale (or even nanoscale) charge writing and ferroelectric programming exploiting an AFM tip have been extensively realized in oxide interface systems. Modulation of superconductivity via creating ferroelastic domains has also been reported [for instance, Y.-Y. Pai, et al., *Phys. Rev. Lett.* 120, 147001 (2018)]. In the present work, the writing and erasing of non-superconducting region are achieved by the creation and elimination of ferroelectric domain with flipped polarization vector, respectively; such domains are characterized by reduced carrier density and boosted impurity scattering rate, indeed, they are most likely to be insulating owing to profound localization effect. As such, the control of superconductivity is actually engendered by introducing an insulating phase at the interface. This, in my opinion, can hardly be regarded as a significant breakthrough on the basis of existing results. I tend to suggest that this manuscript is suitable for a more specialized journal.

Response 1: We thank the referee for acknowledging the substantial efforts we have devoted to addressing the referees' comments and improving the manuscript. However, with due respect, the referee appears to have misunderstood the novelty and central focus of our work—points that we have consistently emphasized in the cover letter, abstract, introduction, conclusion and response letter.

1. Our primary discovery is the emergence of **ferroelectric superconductivity** at the interface of LAO/KTO(111). Examples of this novel superconducting state are exceedingly rare. To date, the only known candidates are $\text{Sr}_{1-x}\text{Ca}_x\text{TiO}_{3-\delta}$ bulk [Ref. 34] and bilayer MoTe_2 [Ref. 35]. The former discovery was published in *Nature Physics* and the latter was in *Nature*, underscoring the scientific significance of this phenomenon and the community's strong interest in it.
2. The discovery of **Interface Ferroelectricity** itself represents an important breakthrough that introduces an entirely new class of ferroelectric materials. While polar interfaces are not uncommon, their electric polarization is typically fixed in space. In stark contrast, the electric polarization at the LAO/KTO(111) interface can be flipped by an external electric field, confirming that it is genuinely ferroelectric. Since interfaces between distinct materials can be

engineered in innumerable combinations, the demonstration of interface ferroelectricity at the interface of two oxides opens a vast space for discovering new ferroelectrics.

3. The technique referenced by the referee concerns only the experimental method used to demonstrate ferroelectric switchability. Although this method has been applied in other contexts, its prior use does not diminish the significance of the phenomena we uncover and therefore is irrelevant to the novelty of our discoveries. The 2018 *PRL* paper mentioned by the referee investigates the one-dimensional nature of superconductivity at the LAO/STO interface, interpreting observations in terms of ferroelastic domains. While we appreciate the referee drawing our attention to this work and have cited it in the revised manuscript, its focus and conclusions are completely different from ours. Needless to say, ferroelasticity and ferroelectricity are fundamentally different phenomena, and the *PRL* paper has nothing to do with interface ferroelectricity or ferroelectric superconductivity.

4. Our findings open exciting avenues for further research of fundamental importance. Theoretically speaking, the breaking of inversion symmetry in a superconducting state can induce mixing between *s*, *p* or *d*-wave pairings, potentially giving rise to novel properties and effects. For ferroelectric superconductivity, little is known and our study represents an initial step towards understanding this new class of superconductivity.

As recognized by the other referees, our discoveries are of broad interest to the general readership. Furthermore, presentations of this work at international conferences have consistently generated strong enthusiasm within the community. We sincerely hope that the referee will reconsider the significance of our findings in light of the above clarifications.

Reviewer #2 (Remarks to the Author):

Comment 2: While the authors have attempted to answer the questions that I asked during the first round of review, two major questions have not been addressed adequately, in my opinion.

Response 2: We have now performed additional experiments that directly address both remaining concerns, providing compelling evidence to clarify these issues. We have added these results to the Supplementary Information.

Comment 3: Regarding the coexistence of ferroelectricity at the superconducting phase: The authors have argued by showing the power dependence of SHG data at 2 K that the sample has polar distortion. However, the resistance at 2 K is 180 K, which clearly establishes that the sample is metallic, not superconducting. Thus, the author's claim of the coexistence of ferroelectricity and superconductivity at the same temperature is not

experimentally demonstrated.

Response 3: As explained in our previous response letter, performing SHG measurements at such a low temperature is technically challenging. Ferroelectric order is expected to strengthen—not weaken—as temperature decreases, owing to the suppression of thermal fluctuations. Indeed, Fig. 2d shows that ferroelectric polarization persists down to 2 K, leaving no reason to expect its disappearance at even lower temperatures. Nevertheless, to unambiguously demonstrate the coexistence of superconductivity and ferroelectricity, we devoted substantial effort to further lowering the base temperature of our SHG measurement setup.

To reduce the heating from the laser beam, we replaced the picosecond laser with a femtosecond laser and lowered the laser power to only 1.5 mW for the measurements shown in Fig. R1, compared with 83 mW in Figs. 2e and 2f. Despite this drastically reduced laser intensity, the SHG signal in Fig. R1 remains significant. With these improvements, the base temperature of the LAO/KTO(111) film reaches 1.62 K—well below its superconducting temperature. During the SHG measurements at 1.62 K, we simultaneously monitored the resistance to ensure that the film stayed in the zero-resistance state, thus ruling out any light-induced transition out of superconductivity.

As shown in Fig. R1a, the resistance of the LAO/KTO(111) sample drops to zero at approximately $T = 1.75$ K (the $R(T)$ curve was measured both in a dilution refrigerator and in the cryostat equipped with SHG optics). Concurrently, an SHG signal is clearly detected at $T = 1.62$ K, when the sample is in the superconducting state. Here the wavelength of the incident laser is 1031 ± 7 nm and a pronounced peak appears near 517 nm, corresponding to the SHG signal. The SHG intensity, $I_{SHG}(T)$, remains roughly the same at elevated temperatures (Figs. R1b and R1c), consistent with Fig. 2d. These results confirm that ferroelectric polarization is robust from room temperature all the way down to the superconducting regime.

Figure R1 | Low-temperature SHG experiments confirm the coexistence of ferroelectricity and superconductivity in LAO/KTO(111). a, Temperature dependence of the resistance $R(T)$, showing the onset of zero-resistance at

~1.75 K. The arrows indicate the temperatures at which SHG measurements were performed. **b**, With an incident laser wavelength of 1031 ± 7 nm, a pronounced peak appears near 517 nm, corresponding to the SHG signal. SHG spectra acquired from $T = 1.62$ to 2.81 K provide clear evidence of ferroelectric order persisting within the superconducting state. **c**, Temperature dependence of the SHG intensity $I_{SHG}(T)$, demonstrating that ferroelectricity remains robust down to 1.62 K and thus coexists with superconductivity.

Comment 4: Role of charge trapping/detrapping under electric field: In spite of acknowledging that this is a possible scenario, the authors have not shown any experimental data to rule out this important possibility.

Response 4: We have conducted new switching experiments that provide evidence against charge trapping as the dominant mechanism (Fig. R2). The critical distinction lies in the hysteretic behavior: charge trapping or detrapping would produce volatile, non-hysteretic resistance changes that track the applied voltage instantaneously. In contrast, our data reveals a completely different, strongly hysteretic behavior (Fig. R2):

1. **Stable bistates above the coercive voltage.** Once the voltage applied to the PFM tip during scanning exceeds the coercive voltage, the LAO/KTO(111) film switches into one of two stable states—low or high resistance—depending solely on the voltage polarity (Fig. R2a).
2. **Non-volatile switching to the high-resistance state.** Starting from the low-resistance state at 0 V, stepwise ramping V_{tip} down to -50 V drives the film to the high-resistance state. When V_{tip} is subsequently ramped back to 0 V, the resistance remains locked, demonstrating clear non-volatile memory (Fig. R2c).
3. **Non-volatile switching to the low-resistance state.** Starting from either a high (Fig. R2d) or low (Fig. R2f) resistance state at 0 V, increasing V_{tip} stepwise to +50 V forces the film into the low-resistance state, which remains stable when V_{tip} is returned to 0 V.
4. **Retention of intermediate states.** When V_{tip} is swept to a value corresponding to an intermediate resistance state (Figs. R2b and R2e), reducing V_{tip} back to 0 V preserves this intermediate state as well.

This pronounced hysteresis, robust retention, and non-volatile memory observed here are hallmark signatures of ferroelectric polarization switching. Charge trapping, by contrast, would cause the resistance to relax upon field removal—behavior absent in Fig. R2. These results therefore provide direct experimental evidence that ferroelectric switching, rather than charge trapping/detrapping, governs the resistance modulation.

Figure R2 | Absence of charge trapping/detrapping in LAO/KTO(111). **a**, The longitudinal resistance manifests clear hysteresis behavior in response to PFM tip writings at 300 K. **b-f**, The tip voltage is ramped from 0 V step-by-step to a designated value V_{max} and then ramped back to 0 V. The sequence of voltage ramping is indicated by the arrows and numberings. The value of V_{max} for panels **b** to **f** are indicated by the labels from “b” to “f” in panel **a**, respectively. Apparently, the hysterical behavior of $R(V_{tip})$ is correlated with ferroelectric hysteresis, rather than charge trapping/detrapping.

Reviewer #3 (Remarks to the Author):

Comment 5: The authors have undertaken a thorough and careful revision of their manuscript, incorporating not only substantial improvements to the text but also presenting several new and compelling experimental findings. Figure R7 (SHG vs T) stands out as particularly intriguing, displaying an unusual and noteworthy behavior. I strongly encourage the authors to further explore this phenomenon in future studies. Given the significant effort invested in strengthening their claims, I believe the paper merits publication in this journal. I also hope it will inspire further research into the role of OxVacs in this and related compounds.

Response 5: We are delighted by the reviewer’s positive and supportive assessment of our revised manuscript. We fully agree that this is a fascinating phenomenon and are grateful for encouragements to explore it further.

Point-by-point response to the referees' comments.

Reviewer #2 (Remarks to the Author):

Comment 1: Based on the new experimental data provided and the comments of other reviewers, I recommend the publication of this manuscript. I appreciate the efforts of the authors in responding to the questions.

Response 1: We are grateful for the referee's recommendation and constructive suggestions during reviewing process.

Reviewer #3 (Remarks to the Author):

Comment 2: In my view, the revised manuscript adequately addresses both my concerns and those of the other reviewers regarding the experimental validation of the paper's claims. The work now appears sound and well-supported in its conclusions. That said, the issue raised by Reviewer 1 concerning the broader or narrower appeal of the topic is, to some extent, subjective. In my opinion, this matter should be decided by the editors rather than the reviewers, given that all three reviewers have accurately outlined the current state-of-the-art.

Response 2: We really appreciate the referee's positive assessment and recognition of the value of our work. It's encouraging for us to keep working on this important topic.